# Wasserstein Iterative Networks
# for Barycenter Estimation

**Alexander Korotin**
Skolkovo Institute of Science and Technology
Artificial Intelligence Research Institute
*Moscow, Russia*
`a.korotin@skoltech.ru`

**Vage Egiazarian**
Skolkovo Institute of Science and Technology
*Moscow, Russia*
`vage.egiazarian@skoltech.ru`

**Lingxiao Li**
Massachusetts Institute of Technology
*Cambridge, Massachusetts, USA*
`lingxiao@mit.edu`

**Evgeny Burnaev**
Skolkovo Institute of Science and Technology
Artificial Intelligence Research Institute
*Moscow, Russia*
`e.burnaev@skoltech.ru`

## Abstract

Wasserstein barycenters have become popular due to their ability to represent the average of probability measures in a geometrically meaningful way. In this paper, we present an algorithm to approximate the Wasserstein-2 barycenters of continuous measures via a generative model. Previous approaches rely on regularization (entropic/quadratic) which introduces bias or on input convex neural networks which are not expressive enough for large-scale tasks. In contrast, our algorithm does not introduce bias and allows using arbitrary neural networks. In addition, based on the celebrity faces dataset, we construct Ave, celeba! *dataset* which can be used for quantitative evaluation of barycenter algorithms by using standard metrics of generative models such as FID.

## 1 Introduction

Wasserstein barycenters [1] provide a geometrically meaningful notion of the average of probability measures based on optimal transport (OT, see [65]). Methods for computing barycenters have been successfully applied to various practical problems. In geometry processing, shape interpolation can be performed by barycenters [58]. In image processing, barycenters are used for color and style translation [49, 44], texture mixing [50] and image interpolation [32, 56]. In language processing, barycenters can be applied to text evaluation [15]. In online learning, barycenters are used for aggregating probabilistic forecasts of experts [31, 47, 27]. In Bayesian inference, the barycenter of subset posteriors converges to the full data posterior [59, 60] allowing efficient computation of full posterior based on barycenters. In reinforcement learning, barycenters are used for uncertainty propagation [41]. Other applications are data augmentation

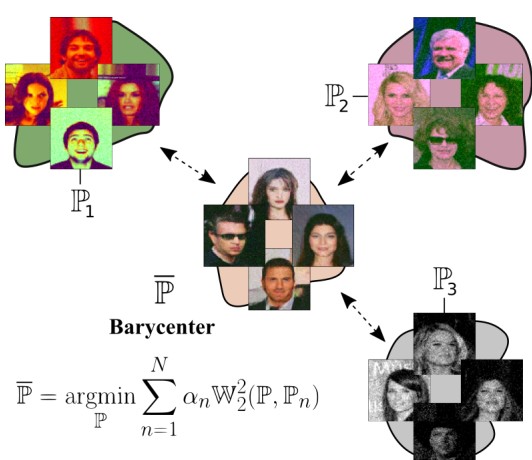

Figure 1: The barycenter w.r.t. weights $(\alpha_1, \alpha_2, \alpha_3) = (\frac{1}{4}, \frac{1}{2}, \frac{1}{4})$ of Ave, Celeba! subsets computed by our Algorithm 1.
The figure shows random samples from the input subsets and generated images from the barycenter.

$$\overline{\mathbb{P}} = \operatorname*{argmin}_{\mathbb{P}} \sum_{n=1}^{N} \alpha_n \mathbb{W}_2^2(\mathbb{P}, \mathbb{P}_n)$$

[8], multivariate density registration [9], distributions alignment [24], domain generalization [39] and adaptation [43], model ensembling [17], averaging of persistence diagrams [63, 7, 6].

The bottleneck of obtaining barycenters is the computational complexity. For *discrete* measures, fast and accurate barycenter algorithms exist for low-dimensional problems; see [48] for a survey. However, discrete methods scale poorly with the number of support points of the barycenter. Consequently, they cannot approximate continuous barycenters well, especially in high dimensions.

Existing continuous barycenter approaches [34, 19, 30] are mostly based on entropic/quadratic regularization or parametrization of Brenier potentials with input-convex neural networks (ICNNs, see [4]). The regularization-based [CR$\mathbb{W}$B] algorithm by [34] recovers a barycenter *biased* from the true one. Algorithms [C$\mathbb{W}_2$B] by [30] and [SC$\mathbb{W}_2$B] by [19] based on ICNNs resolve this issue, see [30, Tables 1-3]. However, despite the growing popularity of ICNNs in OT applications [40, 28, 42], they could be suboptimal architectures according to a recent study [29]. According to the authors, more expressive networks without the convexity constraint outperform ICNNs in practical OT problems.

Furthermore, *evaluation* of barycenter algorithms is challenging due to the limited number of continuous measures with explicitly known barycenters. It can be computed when the input measures are location-scatter (e.g. Gaussians) [3, §4] or 1-dimensional [10, §2.3]. Recent works [34, 30, 19] consider the Gaussian case in dimensions $\leq 256$ for quantitative evaluation. In higher dimensions, the computation of the ground truth barycenter is hard even for the Gaussian case: it involves matrix inversion and square root extraction [2, Algorithm 1] with the cubic complexity in the dimension.

**Contributions.**

- We develop a novel *iterative algorithm* (§4) for estimating Wasserstein-2 barycenters based on the fixed point approach by [3] combined with a neural solver for optimal transport [29]. Unlike predecessors, our algorithm does not introduce bias and allows arbitrary network architectures.
- We construct the *Ave, celeba!* (*ave*raging *celeb*rity faces, §5) dataset consisting of $64 \times 64$ RGB images for large-scale quantitative evaluation of continuous Wasserstein-2 barycenter algorithms. The dataset includes 3 subsets of degraded images of faces (Figure 1). The barycenter of these subsets corresponds to the original clean faces.

Our algorithm is suitable for large-scale Wasserstein-2 barycenters applications. The developed dataset will allow quantitative evaluation of barycenter algorithms at a large scale improving transparency and allowing healthy competition in the optimal transport research.

**Notation.** We work in a Euclidean space $\mathbb{R}^D$ for some $D$. All the integrals are computed over $\mathbb{R}^D$ unless stated otherwise. We denote the set of all Borel probability measures on $\mathbb{R}^D$ with finite second moment by $\mathcal{P}_2(\mathbb{R}^D)$. We use $\mathcal{P}_{2,\text{ac}}(\mathbb{R}^D) \subset \mathcal{P}_2(\mathbb{R}^D)$ to denote the subset of absolutely continuous measures. We denote its subset of measures with positive density by $\mathcal{P}_{2,\text{ac}}^+(\mathbb{R}^D) \subset \mathcal{P}_{2,\text{ac}}(\mathbb{R}^D)$. We denote the set of probability measures on $\mathbb{R}^D \times \mathbb{R}^D$ with marginals $\mathbb{P}$ and $\mathbb{Q}$ by $\Pi(\mathbb{P}, \mathbb{Q})$. For a measurable map $T : \mathbb{R}^D \to \mathbb{R}^D$, we denote the associated push-forward operator by $T\sharp$. For $\phi : \mathbb{R}^D \to \mathbb{R}$, we denote by $\overline{\phi}$ its Legendre-Fenchel transform [20] defined by $\overline{\phi}(y) = \max_{x \in \mathbb{R}^D}[\langle x, y \rangle - \phi(x)]$. Recall that $\overline{\phi}$ is a convex function, even when $\phi$ is not.

## 2 Preliminaries

**Wasserstein-2 distance.** For $\mathbb{P}, \mathbb{Q} \in \mathcal{P}_2(\mathbb{R}^D)$, Monge's *primal* formulation of the squared Wasserstein-2 distance, i.e., OT with *quadratic cost*, is

$$\mathbb{W}_2^2(\mathbb{P}, \mathbb{Q}) \stackrel{\text{def}}{=} \min_{T\sharp\mathbb{P}=\mathbb{Q}} \int \frac{1}{2}\|x - T(x)\|^2 d\mathbb{P}(x), \tag{1}$$

where the minimum is taken over measurable functions (transport maps) $T : \mathbb{R}^D \to \mathbb{R}^D$ mapping $\mathbb{P}$ to $\mathbb{Q}$. The optimal $T^*$ is called the *optimal transport map*. Note that (1) is not symmetric, and this formulation does not allow mass splitting. That is, for some $\mathbb{P}, \mathbb{Q} \in \mathcal{P}_2(\mathbb{R}^D)$, there might be no map $T$ that satisfies $T\sharp\mathbb{P} = \mathbb{Q}$. Thus, [25] proposed the following relaxation:

$$\mathbb{W}_2^2(\mathbb{P}, \mathbb{Q}) \stackrel{\text{def}}{=} \min_{\pi \in \Pi(\mathbb{P}, \mathbb{Q})} \int_{\mathbb{R}^D \times \mathbb{R}^D} \frac{1}{2}\|x - y\|^2 d\pi(x, y), \tag{2}$$

where the minimum is taken over all transport plans $\pi$, i.e., measures on $\mathbb{R}^D \times \mathbb{R}^D$ whose marginals are $\mathbb{P}$ and $\mathbb{Q}$. The optimal $\pi^* \in \Pi(\mathbb{P}, \mathbb{Q})$ is called the *optimal transport plan*. If $\pi^*$ is of the form $[\mathrm{id}, T^*]\sharp\mathbb{P} \in \Pi(\mathbb{P}, \mathbb{Q})$ for some $T^*$, then $T^*$ minimizes (1). The *dual form* [64] of $\mathbb{W}_2^2$ is:

$$\mathbb{W}_2^2(\mathbb{P}, \mathbb{Q}) = \max_{u \oplus v \leq \frac{\|\cdot\|^2}{2}} \left[ \int u(x)d\mathbb{P}(x) + \int v(y)d\mathbb{Q}(y) \right], \tag{3}$$

where the maximum is taken over $u \in \mathcal{L}^1(\mathbb{P})$, $v \in \mathcal{L}^1(\mathbb{Q})$ satisfying $u(x) + v(y) \leq \frac{1}{2}\|x - y\|^2$ for all $x, y \in \mathbb{R}^D$. The functions $u$ and $v$ are called *potentials*. There exist optimal $u^*, v^*$ satisfying $u^* = (v^*)^c$, where $f^c(y) \stackrel{def}{=} \min_{x \in \mathbb{R}^D} \left[ \frac{1}{2}\|x - y\|^2 - f(x) \right]$ is the *c*-transform of $f$. We rewrite (3) as

$$\mathbb{W}_2^2(\mathbb{P}, \mathbb{Q}) = \max_{v} \left[ \int v^c(x)d\mathbb{P}(x) + \int v(y)d\mathbb{Q}(y) \right], \tag{4}$$

where the maximum is taken over all $v \in \mathcal{L}^1(\mathbb{Q})$. It is customary [65, Cases 5.3 & 5.17] to define $u(x) = \frac{1}{2}\|x\|^2 - \psi(x)$ and $v(y) = \frac{1}{2}\|y\|^2 - \phi(y)$. There exist convex optimal $\psi^*$ and $\phi^*$ satisfying $\overline{\psi^*} = \phi^*$ and $\overline{\phi^*} = \psi^*$. If $\mathbb{P} \in \mathcal{P}_{2,ac}(\mathbb{R}^D)$, then the optimal $T^*$ of (1) always exists and can be recovered from the dual solution $u^*$ (or $\psi^*$) of (3): $T^*(x) = x - \nabla u^*(x) = \nabla\psi^*(x)$ [54, Theorem 1.17]. The map $T^*$ is a gradient of a convex function, see the Brenier Theorem [11].

**Wasserstein-2 barycenter**. Let $\mathbb{P}_1, \ldots, \mathbb{P}_N \in \mathcal{P}_{2,ac}(\mathbb{R}^D)$ such that at least one of them has bounded density. Their barycenter w.r.t. weights $\alpha_1, \ldots, \alpha_N$ ($\alpha_n > 0$; $\sum_{n=1}^N \alpha_n = 1$) is given by [1]:

$$\overline{\mathbb{P}} \stackrel{def}{=} \arg\min_{\mathbb{P} \in \mathcal{P}_2(\mathbb{R}^D)} \sum_{n=1}^N \alpha_n \mathbb{W}_2^2(\mathbb{P}_n, \mathbb{P}). \tag{5}$$

The barycenter $\overline{\mathbb{P}}$ exists uniquely and $\overline{\mathbb{P}} \in \mathcal{P}_{2,ac}(\mathbb{R}^D)$. Moreover, its density is bounded [1, Definition 3.6 & Theorem 5.1]. For $n \in \{1, 2, \ldots, N\}$, let $T_{\overline{\mathbb{P}} \to \mathbb{P}_n} = \nabla\psi_n^*$ be the OT maps from $\overline{\mathbb{P}}$ to $\mathbb{P}_n$. The following holds $\overline{\mathbb{P}}$-almost everywhere:

$$\sum_{n=1}^N \alpha_n T_{\overline{\mathbb{P}} \to \mathbb{P}_n}(x) = \sum_{n=1}^N \alpha_n \nabla\psi_n^*(x) = x, \tag{6}$$

see [3, §3]. If $\overline{\mathbb{P}} \in \mathcal{P}_{2,ac}^+(\mathbb{R}^D)$, then (6) holds for every $x \in \mathbb{R}^D$, i.e., $\sum_{n=1}^N \alpha_n \psi_n^*(x) = \frac{\|x\|^2}{2} + c$. We call such convex potentials $\psi_n^*$ *congruent*.

## 3 Related Work

Below we review existing continuous methods for OT. In §3.1, we discuss methods for OT problems (1), (2), (3). In §3.2, we review algorithms that compute barycenters (5).

### 3.1 Continuous OT Solvers for $\mathbb{W}_2$

We use the phrase *OT solver* to denote any method capable of recovering $T^*$ or $u^*$ (or $\psi^*$).

**Primal-form** solvers based on (1) or (2), e.g., [67, 37], parameterize $T^*$ using complicated generative modeling techniques with adversarial losses to handle the pushforward constraint $T\sharp\mathbb{P} = \mathbb{Q}$ in the primal form (1). They depend on careful hyperparameter search and complex optimization [38].

**Dual-form** continuous solvers [21, 55, 46, 62, 28] based on (3) or (4) have straightforward optimization procedures and can be adapted to various tasks without extensive hyperparameter search.

A comprehensive overview and a benchmark of dual-form solvers are given in [29]. According to the evaluation, the best performing OT solver is *reversed maximin solver* $\lfloor$MM:R$\rceil$, a modification of the idea proposed by [46] in the context of Wasserstein-1 GANs [5]. In this paper, we employ this solver as a part of our algorithm. We review it below.

*Reversed Maximin Solver*. In (4), $v^c(x)$ can be expanded through $v$ via the definition of *c*-transform:

$$\max_{v} \int \min_{y \in \mathbb{R}^D} \left[ \frac{\|x - y\|_2^2}{2} - v(y) \right] d\mathbb{P}(x) + \int v(y)d\mathbb{Q}(y) =$$
$$\max_{v} \min_{T} \int \left[ \frac{\|x - T(x)\|_2^2}{2} - v(T(x)) \right] d\mathbb{P}(x) + \int v(y)d\mathbb{Q}(y). \tag{7}$$

In (7), the optimization over $y \in \mathbb{R}^D$ is replaced by the equivalent optimization over functions $T : \mathbb{R}^D \to \mathbb{R}^D$. This is done by the interchanging of integral and minimum, see [51, Theorem 3A].

The key point of this reformulation is that the optimal solution of this maximin problem is given by $(v^*, T^*)$, where $T^*$ is the OT map from $\mathbb{P}$ to $\mathbb{Q}$, see discussion in [29, §2] or [53, §4.1]. In practice, the potential $v : \mathbb{R}^D \to \mathbb{R}$ and the map $T : \mathbb{R}^D \to \mathbb{R}^D$ are parametrized by neural networks $v_\omega, T_\theta$. To train $\theta$ and $\omega$, stochastic gradient ascent/descent (SGAD) over mini-batches from $\mathbb{P}, \mathbb{Q}$ is used.

## 3.2 Algorithms for Continuous $\mathbb{W}_2$ Barycenters

**Variational optimization.** Problem (5) is optimization over probability measures. To estimate $\overline{\mathbb{P}}$, one may employ a generator $G_\xi : \mathbb{R}^H \to \mathbb{R}^D$ with a latent measure $\mathbb{S}$ on $\mathbb{R}^H$ and train $\xi$ by minimizing

$$\sum_{n=1}^{N} \alpha_n \mathbb{W}_2^2(\underbrace{G_\xi \sharp \mathbb{S}}_{\mathbb{P}_\xi}, \mathbb{P}_n) \to \min_\xi. \tag{8}$$

Optimization (8) can be performed by using SGD on random mini-batches from measures $\mathbb{P}_n$ and $\mathbb{S}$. The difference between possible variational algorithms lies in the particular estimation method for $\mathbb{W}_2^2$ terms. To our knowledge, only ICNN-based minimax solver [40] has been used to compute $\mathbb{W}_2^2$ in (8) yielding [SC$\mathbb{W}_2$B] algorithm [19].

**Potential-based optimization.** [34, 30] recover the optimal potentials $\{\psi_n^*, \phi_n^*\}$ for each pair $(\overline{\mathbb{P}}, \mathbb{P}_n)$ via a non-minimax regularized dual formulation. No generative model is needed: the barycenter is recovered by pushing forward measures using gradients of potentials or by barycentric projection. However, the non-trivial choice of the *prior* barycenter distribution is required. Algorithm [CR$\mathbb{W}$B] by [34] use entropic or quadratic regularization and [C$\mathbb{W}_2$B] algorithm by [30] uses ICNNs, congruence and cycle-consistency [28] regularization.

**Other methods.** Recent work [14] combines the variational (8) and potential-based optimization via the $c$-cyclical monotonity regularization. In [16], an algorithm to sample from the continuous Wasserstein barycenter via the gradient flows is proposed.

## 4 Iterative $\mathbb{W}_2$-Barycenter Algorithm

Our proposed algorithm is based on the *fixed point approach* by [3] which we recall in §4.1. In §4.2, we formulate our algorithm for computing Wasserstein-2 barycenters. In §4.3, we show that our algorithm generalizes the variational barycenter approach.

### 4.1 Theoretical Fixed Point Approach

Following [3], we define an operator $\mathcal{H} : \mathcal{P}_{2,ac}(\mathbb{R}^D) \to \mathcal{P}_{2,ac}(\mathbb{R}^D)$ by $\mathcal{H}(\mathbb{P}) = [\sum_{n=1}^{N} \alpha_n T_{\mathbb{P} \to \mathbb{P}_n}] \sharp \mathbb{P}$, where $T_{\mathbb{P} \to \mathbb{P}_n}$ denotes the OT map from $\mathbb{P}$ to $\mathbb{P}_n$. The measure $\mathcal{H}(\mathbb{P})$ obtained by the operator is indeed absolutely continuous, see [3, Theorem 3.1]. According to (6), the barycenter $\overline{\mathbb{P}}$ defined by (5) is a *fixed point* of $\mathcal{H}$, i.e., $\mathcal{H}(\overline{\mathbb{P}}) = \overline{\mathbb{P}}$. This suggests a way to compute $\overline{\mathbb{P}}$ by picking some $\mathbb{P} \in \mathcal{P}_{2,ac}(\mathbb{R}^D)$ and recursively applying $\mathcal{H}$ until convergence. However, there are several **challenges**:

**(a)** A fixed point $\mathbb{P} \in \mathcal{P}_{2,ac}(\mathbb{R}^D)$ satisfying $\mathcal{H}(\mathbb{P}) = \mathbb{P}$ may be not the barycenter [3, Example 3.1]. The situation is analogous to that of the iterative $k$-means algorithm for a different problem – clustering. There may be fixed points which are not globally optimal.

**(b)** The sequence $\{\mathcal{H}^k(\mathbb{P})\}_k$ is tight [3, Theorem 3.6] so it has a subsequence converging in $\mathcal{P}_{2,ac}(\mathbb{R}^D)$, but the entire sequence may not converge. Nevertheless, the value of the objective (5) decreases for $\mathcal{H}^k(\mathbb{P})$ as $k \to \infty$ [3, Prop. 3.3].

**(c)** Efficient parametrization of the evolving measure $\mathcal{H}^k(\mathbb{P})$ is required. Moreover, to get $\mathcal{H}^{k+1}(\mathbb{P})$ from $\mathcal{H}^k(\mathbb{P})$, one needs to compute $N$ optimal transport maps $T_{\mathcal{H}^k(\mathbb{P}) \to \mathbb{P}_n}$ which can be costly.

In [13] and [2], the fixed point approach is considered in the Gaussian case where the sequence $\mathcal{H}^k(\mathbb{P})$ is guaranteed to converge to the unique fixed point – the barycenter. The Gaussian case also makes parameterization **(c)** simple since both measures $\mathbb{P}_n$ and $\mathcal{H}^k(\mathbb{P})$ can be parametrized by means and covariance matrices, and the maps $T_{\mathcal{H}^k(\mathbb{P}) \to \mathbb{P}_n}$ are linear with closed form.

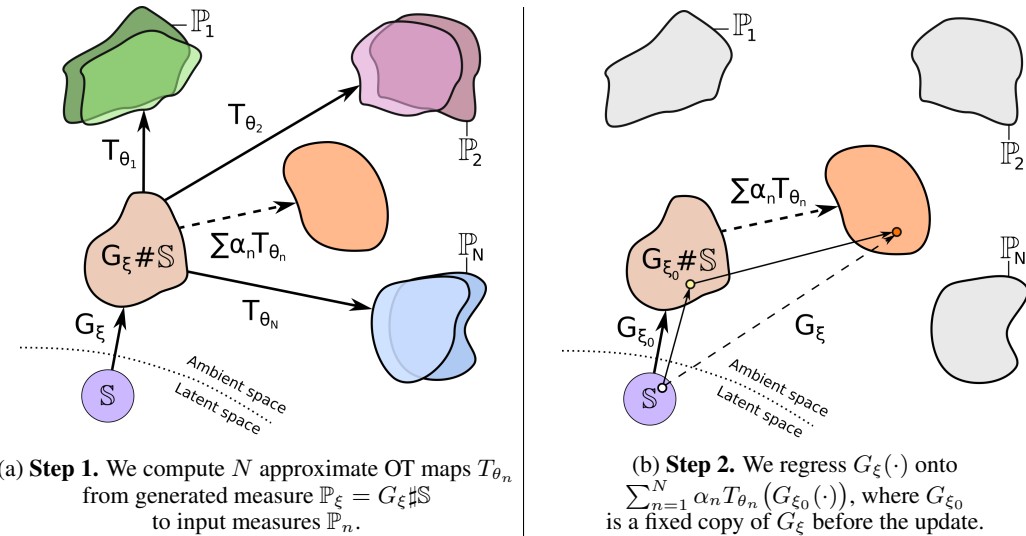

(a) **Step 1.** We compute $N$ approximate OT maps $T_{\theta_n}$ from generated measure $\mathbb{P}_\xi = G_\xi \sharp \mathbb{S}$ to input measures $\mathbb{P}_n$.

(b) **Step 2.** We regress $G_\xi(\cdot)$ onto $\sum_{n=1}^N \alpha_n T_{\theta_n}\big(G_{\xi_0}(\cdot)\big)$, where $G_{\xi_0}$ is a fixed copy of $G_\xi$ before the update.

Figure 2: Our proposed two-step implementation of the fixed-point operator $\mathcal{H}(\cdot)$..

For general continuous measures $\mathbb{P}_n$, it remains an open problem to find sharp conditions on inputs $\mathbb{P}_n$ and the initial measure $\mathbb{P}$ of the fixed-point iteration for the sequence $\{\mathcal{H}^k(\mathbb{P})\}_k$ to converge to the barycenter. In this work, we empirically verify that the fixed point approach works well for the input measures that we consider and for a randomly initialized generative model representing the evolving barycenter (§4.2). We tackle challenge **(c)** and develop a scalable optimization procedure that requires only sample access to $\mathbb{P}_n \in \mathcal{P}_{2,ac}(\mathbb{R}^D)$.

## 4.2 Practical Iterative Optimization Procedure

We employ a generative model to parametrize the evolving measure, i.e., put $\mathbb{P}_\xi = G_\xi \sharp \mathbb{S}$, where $\mathbb{S}$ is a latent measure, e.g., $\mathbb{S} = \mathcal{N}(0, I_H)$, and $G_\xi$ is a neural network $\mathbb{R}^H \to \mathbb{R}^D$ with parameters $\xi$. Our approach to compute the operator $\mathcal{H}(\mathbb{P}_\xi)$ and update $G_\xi$ consists of two steps.

First, we approximately recover $N$ maps $T_{\mathbb{P}_\xi \to \mathbb{P}_n}$ via $\lceil$MM:R$\rceil$ solver, i.e., we use $N$ pairs or networks $\{T_{\theta_n}, v_{\omega_n}\}$ and train them by optimizing (7) with $\mathbb{P} \leftarrow \mathbb{P}_\xi$ and $\mathbb{Q} \leftarrow \mathbb{P}_n$. For each $n = 1, 2, \ldots, N$, we perform SGAD by using batches from $G_\xi \sharp \mathbb{S}$ and $\mathbb{P}_n$ and get $T_{\theta_n} \approx T_{\mathbb{P}_\xi \to \mathbb{P}^n}$.

Second, we update $G_\xi$ to represent $\mathcal{H}(\mathbb{P}_\xi)$ instead of $\mathbb{P}_\xi$. Inspired by [12], we do this via regression. We introduce $G_{\xi_0}$, a fixed copy of $G_\xi$. Next, we regress $G_\xi(\cdot)$ onto $\sum_{n=1}^N \alpha_n T_{\theta_n}\big(G_{\xi_0}(\cdot)\big)$

$$\int_z \boldsymbol{\ell}\bigg(G_\xi(z), \sum_{n=1}^N \alpha_n T_{\theta_n}\big(G_{\xi_0}(z)\big)\bigg) d\mathbb{S}(z) \to \min_\xi$$

by performing SGD on random batches from $\mathbb{S}$, e.g., by using *squared error* $\boldsymbol{\ell}(x, x') \overset{\text{def}}{=} \frac{1}{2}\|x - x'\|^2$. Thus, generator $G_\xi(\cdot)$ becomes close to $\sum_{n=1}^N \alpha_n T_{\theta_n}\big(G_{\xi_0}(\cdot)\big)$ as a function $\mathbb{R}^H \to \mathbb{R}^D$. We get

$$\mathbb{P}_\xi = G_\xi \sharp \mathbb{S} \approx \Big[\sum_{n=1}^N \alpha_n T_{\theta_n}\Big]\sharp\big[G_{\xi_0}\sharp\mathbb{S}\big] = \Big[\sum_{n=1}^N \alpha_n T_{\theta_n}\Big]\sharp\mathbb{P}_{\xi_0} \approx \Big[\sum_{n=1}^N \alpha_n T_{\mathbb{P}_{\xi_0}\to\mathbb{P}_n}\Big]\sharp\mathbb{P}_{\xi_0} = \mathcal{H}(\mathbb{P}_{\xi_0}),$$

i.e., the new generated $G_\xi \sharp \mathbb{S}$ measure approximates $\mathcal{H}(\mathbb{P}_{\xi_0})$.

**Summary.** Our two-step approach iteratively recomputes OT maps $T_{\mathbb{P}_\xi \to \mathbb{P}_n}$ (Figure 2a) and then uses regression to update the generator (Figure 2b). The *optimization procedure* is detailed in Algorithm 1. Note that when fitting OT maps $T_{\mathbb{P}_\xi \to \mathbb{P}^n}$, we start from previously used $\{T_{\theta_n}, v_{\omega_n}\}$ rather than re-initialize them. Empirically, this works better.

## 4.3 Relation to Variational Barycenter Algorithms

We show that our Algorithm 1 reduces to variational approach (§3.2) when the number of generator updates, $K_G$, is equal to $1$. More specifically, we show the equivalence of the gradient update w.r.t.

**Algorithm 1:** Wasserstein Iterative Networks (WIN) for Barycenter Estimation

---

**Input** : latent $\mathbb{S}$ and input $\mathbb{P}_1, \ldots, \mathbb{P}_N$ measures; weights $\alpha_1, \ldots, \alpha_N > 0$ ($\sum_{n=1}^N \alpha_n = 1$);
   number of iters per network: $K_G$, $K_T$, $K_v$; generator $G_\xi : \mathbb{R}^H \to \mathbb{R}^D$;
   mapping networks $T_{\theta_1}, \ldots, T_{\theta_N} : \mathbb{R}^D \to \mathbb{R}^D$; potentials $v_{\omega_1}, \ldots, v_{\omega_N} : \mathbb{R}^D \to \mathbb{R}$;
   regression loss $\ell : \mathbb{R}^D \times \mathbb{R}^D \to \mathbb{R}_+$;
**Output** : generator satisfying $G_\xi \sharp \mathbb{S} \approx \overline{\mathbb{P}}$; OT maps satisfying $T_{\theta_n} \sharp (G_\xi \sharp \mathbb{S}) \approx \mathbb{P}_n$;
**repeat**
 | # *OT solvers update*
 | **for** $n = 1, 2, \ldots, N$ **do**
 | | **for** $k_v = 1, 2, \ldots, K_v$ **do**
 | | | Sample batches $Z \sim \mathbb{S}$, $Y \sim \mathbb{P}_n$; $X \leftarrow G_\xi(Z)$;
 | | | $\mathcal{L}_v \leftarrow \frac{1}{|X|} \sum_{x \in X} v_{\omega_n}\big(T_{\theta_n}(x)\big) - \frac{1}{|Y|} \sum_{y \in Y} v_{\omega_n}(y)$;
 | | | Update $\omega_n$ by using $\frac{\partial \mathcal{L}_v}{\partial \omega_n}$;
 | | | **for** $k_T = 1, 2, \ldots, K_T$ **do**
 | | | | Sample batch $Z \sim \mathbb{S}$; $X \leftarrow G_\xi(Z)$;
 | | | | $\mathcal{L}_T = \frac{1}{|X|} \sum_{x \in X} \big[\frac{1}{2}\|x - T_{\theta_n}(x)\|^2 - v_{\omega_n}\big(T_{\theta_n}(x)\big)\big]$;
 | | | | Update $\theta_n$ by using $\frac{\partial \mathcal{L}_T}{\partial \theta_n}$;
 |
 | # *Generator update (regression)*
 | $G_{\xi_0} \leftarrow \text{copy}(G_\xi)$;
 | **for** $k_G = 1, 2, \ldots, K_G$ **do**
 | | Sample batch $Z \sim \mathbb{S}$;
 | | $\mathcal{L}_G \leftarrow \frac{1}{|Z|} \sum_{z \in Z} \ell\bigg(G_\xi(z), \sum_{n=1}^N \alpha_n T_{\theta_n}\big(G_{\xi_0}(z)\big)\bigg)$;
 | | Update $\xi$ by using $\frac{\partial \mathcal{L}_G}{\partial \xi}$;
**until** *not converged*;

---

parameters $\xi$ of the *generator* in our iterative Algorithm 1 and that of (8). We assume that $\mathbb{W}_2^2$ terms are computed exactly in (8) regardless of the particular OT solver. Similarly, in Algorithm 1, we assume that maps $G_{\xi_0} \sharp \mathbb{S} \to \mathbb{P}_n$ before the generator update are always exact, i.e., $T_{\theta_n} = T_{\mathbb{P}_{\xi_0} \to \mathbb{P}_n}$.

**Lemma 1.** *Assume that $\mathbb{P}_\xi = G_\xi \sharp \mathbb{S} \in \mathcal{P}_{2,ac}(\mathbb{R}^D)$. Consider $K_G = 1$ for the iterative Algorithm 1, i.e., we do a single gradient step regression update per OT solvers' update. Assume that $\ell(x, x') = \frac{1}{2}\|x - x'\|^2$, i.e., the squared loss is used for regression. Then the generator's gradient update in Algorithm 1 is the same as in the variational algorithm:*

$$\frac{\partial}{\partial \xi} \int_z \frac{1}{2} \big\| G_\xi(z) - \sum_{n=1}^N \alpha_n T_{\mathbb{P}_{\xi_0} \to \mathbb{P}_n}\big(G_{\xi_0}(z)\big)\big\|^2 d\mathbb{S}(z) = \frac{\partial}{\partial \xi} \sum_{n=1}^N \alpha_n \mathbb{W}_2^2(G_\xi \sharp \mathbb{S}, \mathbb{P}_n), \qquad (9)$$

*where the derivatives are evaluated at $\xi = \xi_0$.*

We prove the lemma in Appendix A. In practice, we choose $K_G = 50$ as it empirically works better.

## 5 Ave, celeba! Images Dataset

In this section, we develop a generic methodology for building measures with known $\mathbb{W}_2$ barycenter. We then use it to construct Ave, celeba! dataset for quantitative evaluation of barycenter algorithms.

**Key idea.** Consider $\alpha_1, \ldots, \alpha_N > 0$ with $\sum_{n=1}^N \alpha_n = 1$, congruent convex functions $\psi_1, \ldots \psi_N : \mathbb{R}^D \to \mathbb{R}$, and a measure $\mathbb{P} \in \mathcal{P}_{2,ac}^+(\mathbb{R}^D)$ with positive density. Define $\mathbb{P}_n = \nabla \psi_n \sharp \mathbb{P}$. Thanks to Brenier's theorem [11], $\nabla \psi_n$ is the unique OT map from $\mathbb{P}$ to $\mathbb{P}_n$. Since the support of $\mathbb{P}$ is $\mathbb{R}^D$, $\psi_n$ is the unique (up to a constant) dual potential for $(\mathbb{P}, \mathbb{P}_n)$ [61]. Since potentials $\psi_n$'s are congruent, the barycenter $\overline{\mathbb{P}}$ of $\mathbb{P}_n$ w.r.t. weights $\alpha_1, \ldots, \alpha_N$ is $\mathbb{P}$ itself [13, C.2]. If $\psi_n$'s are such that all $\mathbb{P}_n$ are absolutely continuous, then $\mathbb{P} = \overline{\mathbb{P}}$ is the unique barycenter (§2).

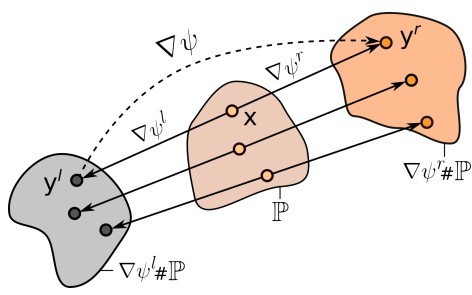
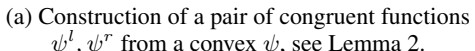
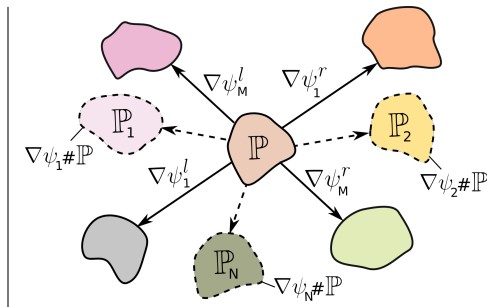

(a) Construction of a pair of congruent functions $\psi^l, \psi^r$ from a convex $\psi$, see Lemma 2.

(b) Construction of $N$ congruent $\psi_n$ as convex combinations of $M$ congruent pairs $(\psi_m^l, \psi_m^r)$, see Lemma 3.

Figure 3: Construction of tuples of congruent functions and production of measures with known $\mathbb{W}_2$ barycenter (§5).

If one obtains $N$ congruent $\psi_n$, then for any $\mathbb{P} \in \mathcal{P}_{2,ac}(\mathbb{R}^D)$, pushforward measures $\mathbb{P}_n = \nabla \psi_n \sharp \mathbb{P}$ can be used as the input measures for the barycenter task. For $\mathbb{P}$ accessible by samples, measures $\mathbb{P}_n$ are also accessible by samples: one may sample $x \sim \mathbb{P}$ and push samples forward by $\nabla \psi_n$.

The challenging part is to construct non-trivial congruent convex functions $\psi_n$. First, we provide a novel method to transform a single convex function $\psi$ into a *pair* $(\psi^l, \psi^r)$ of convex functions satisfying $\alpha \nabla \psi^l(x) + (1-\alpha)\nabla \psi^r(x) = x$ for all $x \in \mathbb{R}^D$ (Lemma 2). Next, we extend the method to generate congruent $N$-*tuples* (Lemma 3).

**Lemma 2** (Constructing congruent pairs)**.** *Let $\psi$ be a strongly convex and $L$-smooth (for some $L > 0$) function. Let $\beta \in (0,1)$. Define $\beta$-left and $\beta$-right functions of $\psi$ by*

$$\psi^l \overset{def}{=} \overline{\beta \frac{\|\cdot\|^2}{2} + (1-\beta)\psi} \qquad and \qquad \psi^r \overset{def}{=} \overline{(1-\beta)\frac{\|\cdot\|^2}{2} + \beta\overline{\psi}}. \qquad (10)$$

*Then $\beta\psi^l(x) + (1-\beta)\psi^r(x) = \frac{\|x\|^2}{2}$ for $x \in \mathbb{R}^D$, i.e., convex functions $\psi^l, \psi^r$ are congruent w.r.t. weights $(\beta, 1-\beta)$. Besides, for all $x \in \mathbb{R}^D$ the gradient $y^l \overset{def}{=} \nabla \psi^l(x)$ can be computed via solving $\beta$-strongly concave optimization:*

$$y^l = \arg\max_{y \in \mathbb{R}} \left( \langle x, y \rangle - \beta \frac{\|y\|^2}{2} - (1-\beta)\psi(y) \right). \qquad (11)$$

*In turn, the value $y^r \overset{def}{=} \nabla \psi^r(x)$ is given by $y^r = \nabla \psi(y^l)$.*

The proof is given in Appendix A. We visualize the idea of our Lemma 2 in Figure 3a. Thanks to Lemma 2, any analytically known convex $\psi$, e.g., an ICNN, can be used to produce a congruent pair $\psi^l, \psi^r$. To compute the gradient maps, optimization (11) can be solved by convex optimization tools with $\nabla \psi$ computed by automatic differentiation.

**Lemma 3** (Constructing $N$ congruent functions.)**.** *Let $\psi_1^0, \ldots, \psi_M^0$ be convex functions, $\beta_1, \ldots, \beta_M \in (0,1)$ and $\psi_m^l, \psi_m^r$ be $\beta_m$-left, $\beta_m$-right functions for $\psi_m^0$ respectively. Let $\gamma^l, \gamma^r \in \mathbb{R}^{N \times M}$ be two rectangular matrices with non-negative elements and the sum of elements in each column equals to 1. Let $w_1, \ldots, w_M > 0$ satisfy $\sum_{m=1}^M w_m = 1$. For $n = 1, \ldots, N$ define*

$$\psi_n(x) \overset{def}{=} \frac{\sum_{m=1}^M w_m \left[ \beta_m \gamma_{nm}^l \psi_m^l(x) + (1-\beta_m)\gamma_{nm}^r \psi_m^r(x) \right]}{\sum_{m=1}^M w_m \left[ \beta_m \gamma_{nm}^l + (1-\beta_m)\gamma_{nm}^r \right]}. \qquad (12)$$

*Then $\psi_1, \ldots, \psi_N$ are congruent w.r.t. weights $\alpha_n \overset{def}{=} \sum_{m=1}^M w_m \left[ \beta_m \gamma_{nm}^l + (1-\beta_m)\gamma_{nm}^r \right]$.*

We prove Lemma 3 in Appendix A. We visualize the idea of our Lemma 3 in Figure 3b. The lemma provides an elegant way to create $N \geq 2$ congruent functions from convex linear combinations of functions in given congruent pairs $(\psi_m^l, \psi_m^r)$. Gradients $\nabla \psi_n$ of these functions are respective linear combinations of gradients $\nabla \psi_m^l$ and $\nabla \psi_m^r$.

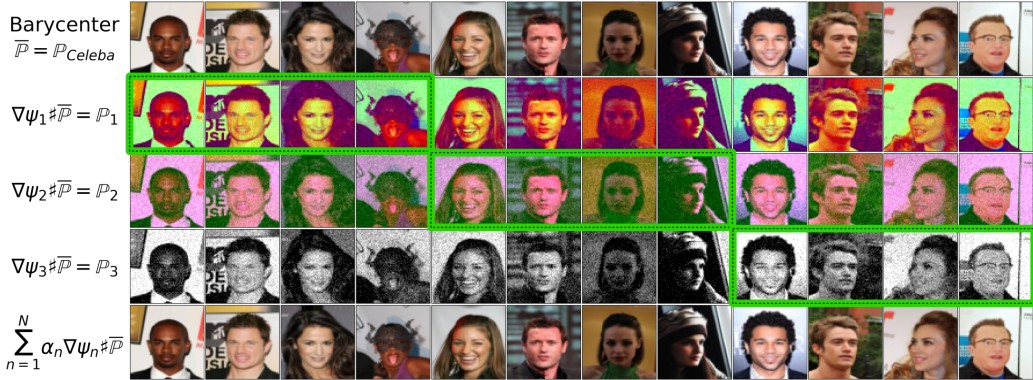

Figure 4: The production of **Ave, celeba! dataset**. The 1st line shows images $x \sim \mathbb{P}_{\text{Celeba}}$. Each of 3 next lines shows OT maps $\nabla\psi_n(x) \sim \nabla\psi_n \sharp \mathbb{P}_{\text{Celeba}} = \mathbb{P}_n$ to constructed measures $\mathbb{P}_n$. Their barycenter w.r.t. $(\alpha_1, \alpha_2, \alpha_3) = (\frac{1}{4}, \frac{1}{2}, \frac{1}{4})$ is $\mathbb{P}_{\text{Celeba}}$. The last line shows congruence of $\psi_n$, i.e., $\sum_{n=1}^{N} \alpha_n \nabla\psi_n(x) \equiv x$. Samples **in green boxes** are included to dataset.

**Dataset creation.** We use CelebA $64 \times 64$ faces dataset [36] as the basis for our Ave, celeba! dataset. We assume that CelebA dataset is an empirical sample from the continuous measure $\mathbb{P}_{\text{Celeba}} \in \mathcal{P}_{2,\text{ac}}^{+}(\mathbb{R}^{3 \times 64 \times 64})$ which we put to be the barycenter in our design, i.e., $\overline{\mathbb{P}} = \mathbb{P}_{\text{Celeba}}$. We construct diffirentiable congruent $\psi_n$ with bijective gradients that produce $\mathbb{P}_n = \nabla\psi_n \sharp \overline{\mathbb{P}} \in \mathcal{P}_{2,\text{ac}}^{+}(\mathbb{R}^{3 \times 64 \times 64})$ whose unique barycenter is $\mathbb{P}_{\text{Celeba}}$. In Lemma 3, we set $N = 3$, $M = 2$, $\beta_1 = \beta_2 = \frac{1}{2}$, $w_1 = w_2 = \frac{1}{2}$ and

$$(\gamma^l)^\top = \begin{pmatrix} 1 & 0 & 0 \\ 0 & 1 & 0 \end{pmatrix}, \qquad (\gamma^r)^\top = \begin{pmatrix} 0 & 1 & 0 \\ 0 & 0 & 1 \end{pmatrix}$$

which yields weights $(\alpha_1, \alpha_2, \alpha_3) = (\frac{1}{4}, \frac{1}{2}, \frac{1}{4})$ We choose the constants above manually to make sure the final produced measures $\mathbb{P}_n$ are visually distinguishable. We use $\psi_m^0(x) = \text{ICNN}_m\big(s_m(\sigma_m(d_m(x)))\big) + \lambda \frac{\|x\|^2}{2}$ as convex functions, where ICNNs have ConvICNN64 architecture [29, Appendix B.1], $\sigma_1, \sigma_2$ are random permutations of pixels and channels, $s_1, s_2$ are axis-wise random reflections, $\lambda = \frac{1}{100}$. In both functions, $d_m$ is a de-colorization transform which sets R, G, B channels of each pixel to $(\frac{7}{10}\text{R} + \frac{1}{25}\text{G} + \frac{13}{50}\text{B})$ for $\psi_1^0$ and $\frac{1}{3}(\text{R} + \text{G} + \text{B})$ for $\psi_2^0$. The weights of ICNNs are initialized by the pre-trained potentials of $\mathbb{W}_2^2$ "Early" transport benchmark which map blurry faces to the clean ones [29, §4.1]. All the implementation details are given in Appendix B.1.

Finally, to create *Ave, celeba!* dataset, we randomly split the images dataset into 3 equal parts containing $\approx 67\text{K}$ samples, and map each part to respective measure $\mathbb{P}_n = \nabla\psi_n \sharp \mathbb{P}_{\text{Celeba}}$ by $\nabla\psi_n$. Resulting $3 \times 67\text{K}$ samples form the dataset consisting of 3 parts. We show the samples in Figure 4. The samples from the respective parts are in green boxes.

## 6 Evaluation

The code[1] is written on the PyTorch and includes the script for producing Ave, celeba! dataset. The experiments are conducted on 4×GPU GTX 1080ti. The details are given in Appendix B.

### 6.1 Evaluation on Ave, celeba! Dataset

We evaluate our *iterative* algorithm 1 and a recent state-of-the-art *variational* [SC$\mathbb{W}_2$B] by [19] on Ave, celeba! dataset. Both algorithms use a generative model $\mathbb{P}_\xi = G_\xi \sharp \mathbb{S}$ for the barycenter and yield approximate maps $\widehat{T}_{\mathbb{P}_\xi \to \mathbb{P}_n}$ to input measures. In our case, the maps are neural networks $T_{\theta_n}$, while in [SC$\mathbb{W}_2$B] they are gradients of ICNNs. The barycenters of Ave, celeba! fitted by our algorithm and [SC$\mathbb{W}_2$B] are shown in Figures 5a and 5b respectively. Recall the ground truth barycenter is $\mathbb{P}_{\text{Celeba}}$. Thus, for quantitative evaluation we use FID score [23] computed on 200K generated samples w.r.t. the original CelebA dataset, see Table 1. Our method *drastically* outperforms [SC$\mathbb{W}_2$B]. Presumably, this is due to the latter using ICNNs which do not provide sufficient performance.

---

[1]https://github.com/iamalexkorotin/WassersteinIterativeNetworks

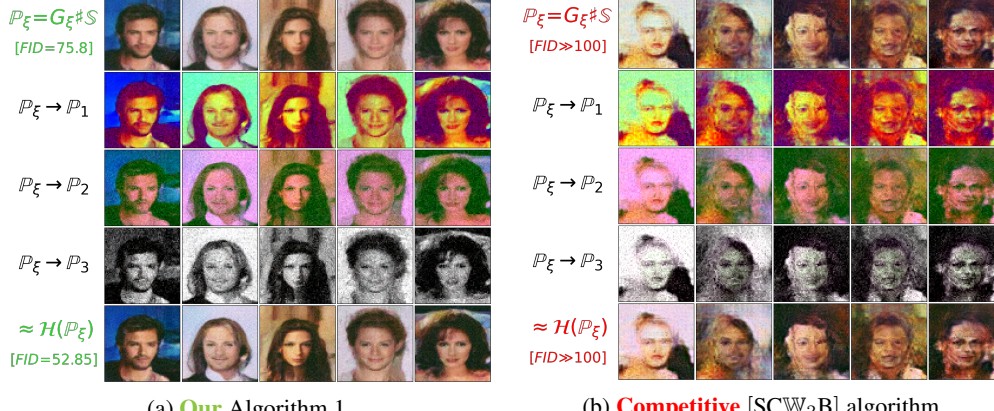

(a) **Our** Algorithm 1.  (b) **Competitive** [SCW$_2$B] algorithm.

Figure 5: The barycenter and maps to input measures estimated by barycenter algorithms. The 1st line shows generated samples $\mathbb{P}_\xi = G_\xi \sharp \mathbb{S} \approx \mathbb{P}_{\text{Celeba}}$. Lines 2-4 show maps $\widehat{T}_{\mathbb{P}_\xi \to \mathbb{P}_n}$. The last line shows the average map $\sum_{n=1}^N \alpha_n \widehat{T}_{\mathbb{P}_\xi \to \mathbb{P}_n}$.

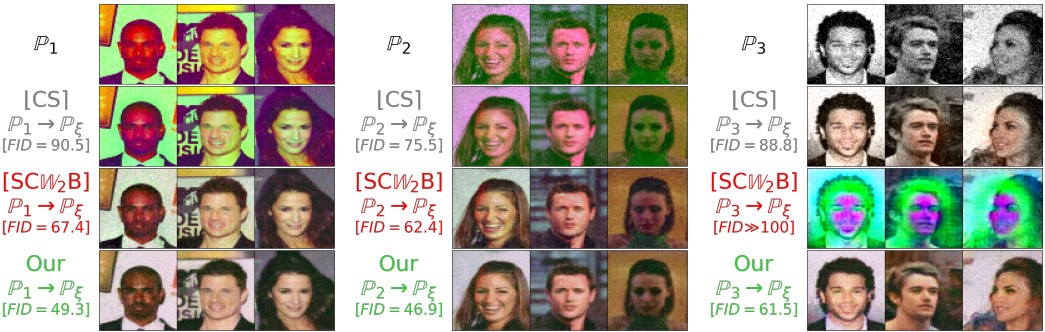

(a) Maps from $\mathbb{P}_1$ to the barycenter.  (b) Maps from $\mathbb{P}_2$ to the barycenter.  (c) Maps from $\mathbb{P}_3$ to the barycenter.

Figure 6: Maps from inputs $\mathbb{P}_n$ to the barycenter $\overline{\mathbb{P}}$ estimated by the barycenter algorithms in view. For comparison with the original barycenter images, the faces are the same as in Figure 4.

| Method | | FID↓ |
|---|---|---|
| [SCW$_2$B] | $G_\xi(z)$ | 156.3 |
| | $\sum_{n=1}^N \alpha_n \widehat{T}_{\mathbb{P}_\xi \to \mathbb{P}_n}(G_\xi(z))$ | 152.1 |
| **Ours** | $G_\xi(z)$ | 75.8 |
| | $\sum_{n=1}^N \alpha_n \widehat{T}_{\mathbb{P}_\xi \to \mathbb{P}_n}(G_\xi(z))$ | 52.85 |

Table 1: FID scores of images from the learned barycenter.

| Method | | FID↓ | | |
|---|---|---|---|---|
| | | $n=1$ | $n=2$ | $n=3$ |
| $\lfloor$CS$\rceil$ | $y + (\overline{\mu} - \mu_n)$ | 90.5 | 75.5 | 88.8 |
| [SCW$_2$B] | $\widehat{T}_{\mathbb{P}_n \to \mathbb{P}_\xi}(y)$ | 67.4 | 62.4 | 319.62 |
| **Ours** | $\widehat{T}_{\mathbb{P}_n \to \mathbb{P}_\xi}(y)$ | 49.3 | 46.9 | 61.5 |

Table 2: FID scores of images mapped from inputs $\mathbb{P}_n$.

Additionally, we evaluate to which extent the algorithms allow to recover the inverse OT maps $T_{\mathbb{P}_n \to \mathbb{P}_\xi}$ from inputs $\mathbb{P}_n$ to the barycenter $\mathbb{P}_\xi \approx \overline{\mathbb{P}}$. In [SCW$_2$B], these maps are computed during training. Our algorithm does not compute them. Thus, we separately fit the inverse maps after main training by using $\lfloor$MM:R$\rceil$ solver between each input $\mathbb{P}_n$ and learned $\mathbb{P}_\xi$ (Algorithm 2 of Appendix 2). The inverse maps are given in Figure 6; their FID scores – in Table 2. Here we add an additional *constant shift* $\lceil$CS$\rfloor$ baseline which simply shifts the mean of input $\mathbb{P}_n$ to the mean $\overline{\mu}$ of $\overline{\mathbb{P}}$. The vector $\overline{\mu}$ is given by $\sum_{n=1}^N \alpha_n \mu_n$, where $\mu_n$ is the mean of $\mathbb{P}_n$ [3]. We estimate $\overline{\mu}$ from samples $y \sim \mathbb{P}_n$.

## 6.2 Additional Experimental Results

**Different domains.** To stress-test our algorithm 1, we compute the barycenters w.r.t. $(\alpha_1, \alpha_2, \alpha_3) = (\frac{1}{3}, \frac{1}{3}, \frac{1}{3})$ of notably different datasets: 50K Shoes [68], 138K Amazon Handbags and 90K Fruits [45]. All the images are rescaled to $64 \times 64$. The *ground gruth barycenter is unknown*, but one may imagine what it looks like. Due to (6), each barycenter image is a *pixel-wise average* of a shoe, a handbag and a fruit, which is supported by our result shown in Figure 7. In the same figure, we also show the maps between datasets *through* the barycenter (Figures 7b, 7c, 7d): this allows generation of images in other categories with styles similar to a given image. For instance, in Figure 7d, we

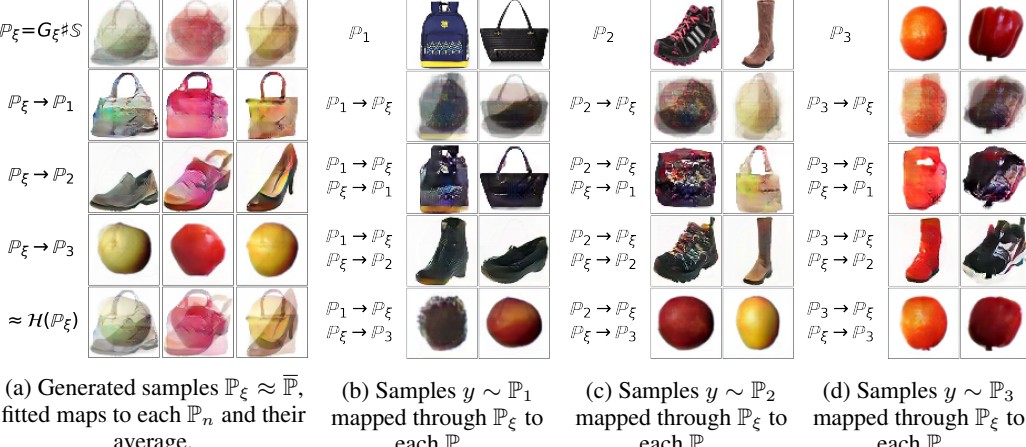

(a) Generated samples $\mathbb{P}_\xi \approx \overline{\mathbb{P}}$, fitted maps to each $\mathbb{P}_n$ and their average.

(b) Samples $y \sim \mathbb{P}_1$ mapped through $\mathbb{P}_\xi$ to each $\mathbb{P}_n$.

(c) Samples $y \sim \mathbb{P}_2$ mapped through $\mathbb{P}_\xi$ to each $\mathbb{P}_n$.

(d) Samples $y \sim \mathbb{P}_3$ mapped through $\mathbb{P}_\xi$ to each $\mathbb{P}_n$.

Figure 7: The barycenter of Handbags, Shoes, Fruit ($64 \times 64$) datasets fitted by **our** Algorithm 1. We give the results of **competitive** [SCW$_2$B] in Appendix B.4.

generate an orange bag and an orange shoe by pushing the image of an orange through the barycenter. We provide more examples in Figure 15 of Appendix C.5.

**Extra results.** In Appendix C.1, we compute the barycenters of toy 2D dsitributions. In Appendix C.2, we provide *quantitative* results for computing barycenters in the Gaussian case. In Appendix C.3, we test how our algorithm works as a *generative model* on the original CelebA dataset, i.e., when $N = 1$ and $\mathbb{P}_1 = \mathbb{P}_{\text{Celeba}}$. We show that in this case it achieves FID scores comparable to recent WGAN models. In Appendix C.4, similar to [19], we compute barycenters of digit classes 0/1 of $32 \times 32$ grayscale MNIST [33]. We also test our algorithm on FashionMNIST [66] **10 classes** dataset.

## 7 Discussion

**Potential impact (algorithm).** We present a scalable barycenter algorithm based on fixed-point iterations with many application prospects. For instance, in medical imaging, MRI is often acquired at multiple sites where the overlap of information (imaging, genetic, diagnosis) between any two sites is limited. Consequently, the data on each site may be biased and can cause generalizability and robustness issues when training models. The developed algorithm could help aggregate data from multiple sites and overcome the distributional shift issue across sites.

**Potential impact (dataset).** There is no high-dimensional dataset for the barycenter problem except for location-scattered cases (e.g. Gaussians), where the transport maps are always linear. Hence our proposed dataset fills an important gap, thereby allowing quantitative evaluation of future related methods. We expect our Ave, celeba! to become a standard dataset for evaluating continuous barycenter algorithms. In addition, we describe a generic recipe (§5) to produce new datasets.

**Limitations (algorithm).** In our algorithm, the evolving measure $\mathbb{P}_\xi$ is not guaranteed to be continuous, while it is continuous in the underlying fixed point approach. To enforce the absolute continuity of $\mathbb{P}_\xi = G_\xi \sharp \mathbb{S}$, one may use an invertible network [18] for $G_\xi$ and an absolutely continuous latent measure $\mathbb{S}$ in a latent space of dimension $H = D$. However, our results suggest this is unnecessary in practice — common GANs approaches also assume $H \ll D$. During the fixed-point iterations, the barycenter objective (5) decreases. However, there is no guarantee that the sequence of measures converges to a fixed point which is the barycenter. Identifying the precise conditions on the input measures and the initial point is an important future direction. Besides, our algorithm does not recover inverse OT maps $T_{\mathbb{P}_n \to \overline{\mathbb{P}}}$; we compute them with an OT solver as a follow-up. To avoid this second step, one may consider using invertible neural nets [18] to parametrize maps $T_{\theta_n}$ in our Algorithm 1.

**Limitations (dataset).** To create Ave, celeba! dataset (§5), we compose ICNNs with decolorization, random reflections and permutations to simulate degraded images. It is unclear how to produce other practically interesting effects via ICNNs. It also remains an open question on how to better hide the information of the barycenter image in the constructed marginal measures. Studying these questions is an interesting future direction that can inspire benchmarking other OT problems.

ACKNOWLEDGEMENTS. E. Burnaev was supported by the Russian Foundation for Basic Research grant 21-51-12005 NNIO_a. A portion of this project was funded by the Skolkovo Institute of Science and Technology as part of the Skoltech NGP Program and funds were received by the Massachusetts Institute of Technology prior to September 1, 2022. Neither Mr. Li, nor any other MIT personnel, contributed to any substantive or artistic alteration or enhancement of this publication after August 31, 2022.

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
