# A Proofs

First, we recall basic properties of convex conjugate functions that we rely on in our proofs. Let $\psi : \mathbb{R}^D \to \mathbb{R}$ be a convex function and $\overline{\psi}$ be its convex conjugate. From the definition of $\overline{\psi}$, we obtain

$$\psi(x) + \overline{\psi}(y) \geq \langle x, y \rangle$$

for all $x, y \in \mathbb{R}^D$. Assume that $\psi$ is differentiable and has an invertible gradient $\nabla\psi : \mathbb{R}^D \to \mathbb{R}^D$. The latter condition holds, e.g., for strongly convex functions. From the convexity of $\psi$, we derive

$$x = \arg\max_{x \in \mathbb{R}^D}[\langle x, y \rangle - \psi(x)] \Leftrightarrow y = \nabla\psi(x) \Leftrightarrow x = (\nabla\psi)^{-1}(y),$$

which yields

$$\overline{\psi}(y) = \langle (\nabla\psi)^{-1}(x), x \rangle - \psi\big((\nabla\psi)^{-1}(x)\big).$$

In particular, the strict equality $\psi(x) + \overline{\psi}(y) = \langle x, y \rangle$ holds if and only if $y = \nabla\psi(x)$. By applying the same logic to $\overline{\psi}$, we obtain $(\nabla\overline{\psi})^{-1} = \nabla\psi$ and $(\nabla\psi)^{-1} = \nabla\overline{\psi}$, i.e., the gradients of conjugate functions are mutually inverse.

## A.1 Proof of Lemma 1

*Proof.* For each $n = 1, 2, \ldots, N$ we perform the following evaluation:

$$\frac{\partial}{\partial\xi}\mathbb{W}_2^2(G_\xi\sharp\mathbb{S}, \mathbb{P}_n) = \int_z \mathbf{J}_\xi G_\xi(z)^T \nabla u_n^*\big(G_\xi(z)\big) d\mathbb{S}(z), \tag{13}$$

$$\int_z \mathbf{J}_\xi G_\xi(z)^T \Big(G_\xi(z) - T_{\mathbb{P}_\xi \to \mathbb{P}_n}\big(G_\xi(z)\big)\Big) d\mathbb{S}(z), \tag{14}$$

where $u_n^*$ is the optimal dual potential for $\mathbb{P}_\xi = G_\xi\sharp\mathbb{S}$ and $\mathbb{P}_n$. Equation (13) follows from [22, Equation 3]. Equation (14) follows from the property $\nabla u_n^*(x) = x - T_{\mathbb{P}_\xi \to \mathbb{P}_n}(x)$ connecting dual potentials and OT maps. We sum (14) for $n = 1, \ldots, N$ w.r.t. weights $\alpha_n$ with $\xi = \xi_0$ and obtain

$$\frac{\partial}{\partial\xi}\sum_{n=1}^N \alpha_n \mathbb{W}_2^2(G_\xi\sharp\mathbb{S}, \mathbb{P}_n) = \int_z \mathbf{J}_\xi G_{\xi_0}(z)^T \Big(G_{\xi_0}(z) - \sum_{n=1}^N \alpha_n T_{\mathbb{P}_{\xi_0} \to \mathbb{P}_n}\big(G_{\xi_0}(z)\big)\Big) d\mathbb{S}(z). \tag{15}$$

Note that (15) exactly matches the derivative of the left-hand side of (9) evaluated at $\xi = \xi_0$. $\qquad\square$

## A.2 Proof of Lemma 2

*Proof.* First, we prove the congruence, i.e., $\beta\psi^l(x) + (1 - \beta)\psi^r(x) = \frac{\|x\|^2}{2}$ for all $x \in \mathbb{R}^D$.

$$\beta\psi^l(x) + (1 - \beta)\psi^r(x) =$$

$$\beta \max_{y_1 \in \mathbb{R}^D}\big[\langle x, y_1 \rangle - \overline{\psi^l}(y_1)\big] + (1 - \beta)\max_{y_2 \in \mathbb{R}^D}\big[\langle x, y_2 \rangle - \overline{\psi^r}(y_2)\big] = \tag{16}$$

$$\beta \max_{y_1 \in \mathbb{R}^D}\Big[\langle x, y_1 \rangle - \beta\frac{\|y_1\|^2}{2} - (1 - \beta)\psi(y_1)\Big] +$$

$$(1 - \beta)\max_{y_2 \in \mathbb{R}^D}\Big[\langle x, y_2 \rangle - (1 - \beta)\frac{\|y_2\|^2}{2} - \beta\overline{\psi}(x)\Big] =$$

$$\max_{y_1, y_2 \in \mathbb{R}^D}\Big[\langle x, \beta y_1 + (1 - \beta)y_2 \rangle - \beta^2\frac{\|y_1\|^2}{2} - (1 - \beta)^2\frac{\|y_2\|^2}{2} - \beta(1 - \beta)(\psi(y_1) + \overline{\psi}(y_2))\Big] \leq \tag{17}$$

$$\max_{y_1, y_2 \in \mathbb{R}^D}\Big[\langle x, \beta y_1 + (1 - \beta)y_2 \rangle - \beta^2\frac{\|y_1\|^2}{2} - (1 - \beta)^2\frac{\|y_2\|^2}{2} - \beta(1 - \beta)\langle y_1, y_2 \rangle\Big] =$$

$$\max_{y_1, y_2 \in \mathbb{R}^D}\frac{\|x\|^2}{2} - \frac{1}{2}\|x - (\beta y_1 + (1 - \beta)y_2)\|^2 \leq \frac{\|x\|^2}{2}. \tag{18}$$

First, we substitute $(y_1, y_2) = (y^l, \nabla\psi(y^l))$. For this pair, $x = \nabla\overline{\psi^l}(y^l) = \beta y^l + (1 - \beta)\nabla\psi(y^l)$, which results in $x = \beta y_1 + (1 - \beta)y_2$. Moreover, since $y_2 = \nabla\psi(y_1)$, we have $\psi(y_1) + \overline{\psi}(y_2) =$

$\langle y_1, y_2 \rangle$. As the consequence, both inequalities (17) and (18) turn to strict equalities yielding congruence of $\psi^l, \psi^r$. From (10), the smoothness and strong convexity of $\psi$ imply that $\psi^l$ and $\psi^r$ are smooth. Consequently, $\overline{\psi^l}$ and $\overline{\psi^l}$ are strongly convex. Thus, the maximizer of (16) is unique. We know the maximum of (16) is attained at $(y_1, y_2) = (\nabla \psi^l(x), \nabla \psi^r(x)) = (y^l, y^r)$. We conclude $(y^l, y^r) = (y^l, \nabla \psi(y^l))$, i.e., $y^r = \nabla \psi(y^l)$. Finally, $y^l = \nabla \psi^l(x) \Leftrightarrow x = \nabla \overline{\psi^l}(y^l) \Leftrightarrow y^l = \max_{y \in \mathbb{R}^D} \left[ \langle x, y \rangle - \overline{\psi^l}(y) \right]$, which matches (11). $\square$

## A.3 Proof of Lemma 3

*Proof.* First, we check that $\sum_{n=1}^{N} \alpha_n$ indeed equals 1:

$$\sum_{n=1}^{N} \alpha_n = \sum_{n=1}^{N} \sum_{m=1}^{M} w_m \left[ \beta_m \gamma_{nm}^l + (1 - \beta_m) \gamma_{nm}^r \right] =$$

$$\sum_{m=1}^{M} \left[ w_m \beta_m \underbrace{\sum_{n=1}^{N} \gamma_{nm}^l}_{=1} \right] + \sum_{m=1}^{M} \left[ w_m (1 - \beta_m) \underbrace{\sum_{n=1}^{N} \gamma_{nm}^r}_{=1} \right] =$$

$$\sum_{m=1}^{M} w_m \beta_m + \sum_{m=1}^{M} w_m (1 - \beta_m) = \sum_{m=1}^{M} w_m \left( \beta_m + (1 - \beta_m) \right) = \sum_{m=1}^{M} w_m = 1. \qquad (19)$$

Next, we check that $\psi_1, \ldots, \psi_N$ are congruent w.r.t. weights $\alpha_1, \ldots, \alpha_N$:

$$\sum_{n=1}^{N} \alpha_n \psi_n(x) = \sum_{n=1}^{N} \sum_{m=1}^{M} w_m \left[ \beta_m \gamma_{nm}^l \cdot \psi_m^l(x) + (1 - \beta_m) \gamma_{nm}^r \cdot \psi_m^r(x) \right] =$$

$$\sum_{m=1}^{M} \left[ w_m \beta_m \psi_m^l(x) \underbrace{\sum_{n=1}^{N} \gamma_{nm}^l}_{=1} \right] + \sum_{m=1}^{M} \left[ w_m (1 - \beta_m) \psi_m^r(x) \underbrace{\sum_{n=1}^{N} \gamma_{nm}^r}_{=1} \right] =$$

$$\sum_{m=1}^{M} \left[ w_m \underbrace{\left( \beta_m \psi_m^l(x) + (1 - \beta_m) \psi^r(x) \right)}_{= \frac{\|x\|^2}{2}} \right] = \sum_{m=1}^{M} w_m \frac{\|x\|^2}{2} = \frac{\|x\|^2}{2}.$$

$\square$

# B Experimental Details

## B.1 Ave, celeba! Dataset Creation

The initialization of random permutations $\sigma_m$ and reflections $s_m$ (for $m = 1, 2$) as well as the random split of CelebA dataset into 3 parts (each containing $\approx 67K$ images) are *hardcoded* in our provided script for producing Ave, celeba! dataset. To initialize $\text{ICNN}_m$ (for $m = 1, 2$), we use use ConvICNN64 [29, Appendix B.1] checkpoints `Early_v1_conj.pt`, `Early_v2_conj.pt` from the official Wasserstein-2 benchmark repository[2].

We rescale Celeba images to $64 \times 64$ by using `imresize` from `scipy.misc`. To create empirical samples from input distributions $\mathbb{P}_n$ by using the rescaled CelebA dataset, we compute the gradient maps $\nabla \psi_n(x)$ ($n = 1, 2, 3$) in Lemma 3 for images $x$ in the CelebA dataset. This computation implies computing gradient maps $\nabla \psi_m^l(x)$ and $\nabla \psi_m^r(x)$ for each base function $\psi_m^0$ ($m = 1, 2$) and summing them with respective coefficients (12). Following our Lemma 2, we compute $y_m^l \stackrel{\text{def}}{=} \nabla \psi_m^l(x)$ by solving a concave optimization problem (11) over the space of images. We solve this problem with the gradient descent. We use Adam optimizer [26] with default betas, $lr = 2 \cdot 10^{-2}$ and do 1000 gradient steps. To speed up the computation, we simultaneously solve the problem for a batch of 256 images $x$ from CelebA dataset. Then we compute $y^r \stackrel{\text{def}}{=} \nabla \psi_m^r(x)$ as $y^r = \nabla \psi_m(y^l)$ (Lemma 2).

---

[2] https://github.com/iamalexkorotin/Wasserstein2Benchmark

**Computational complexity.** Producing Ave, celeba! takes about $1, 5$ days on a GPU GTX 1080 ti.

## B.2 Hyperparameters (Algorithm 1, Main Training)

We provide the hyperparameters of all the experiments with algorithm 1 in Table 3. The column **total iters** shows the sum of gradient steps over generator $G_\xi$ and each of $N$ potentials $v_{\omega_n}$ in OT solvers.

**Optimization.** We use Adam optimizer with the default betas. During training, we decrease the learning rates of the generator $G_\xi$ and each potential $v_{\omega_n}$ every 10K steps of their optimizers. In the Gaussian case, we use a single GPU GTX 1080ti. In all other cases we split the batch over 4×GPU GTX 1080ti (`nn.DataParallel` in PyTorch).

**Neural Network Architectures.** In the Gaussian case, we use In the evaluation in the Gaussian case, we use sequential fully-connected neural networks with ReLU activations for the generator $G_\xi : \mathbb{R}^D \to \mathbb{R}^D$, potentials $v_{\omega_n} : \mathbb{R}^D \to \mathbb{R}$ and transport maps $T_{\theta_n} : \mathbb{R}^D \to \mathbb{R}^D$. For all the networks the sizes of hidden layers are:

$$[\max(100, 2D), \max(100, 2D), \max(100, 2D)].$$

Working with images, we use the ResNet[3] generator and discriminator architectures of WGAN-QC [35] for our generator $G_\xi$ and potentials $v_{\omega_n}$ respectively. As the maps $T_{\theta_n}$, we use U-Net [4] [52].

**Generator regression loss.** In the Gaussian case and experiments with grayscale images (MNIST, FashionMNIST), we use mean squared loss for generator regression. In other experiments, we use the perceptual mean squared loss based on the features of the pre-trained VGG-16 network [57]. The loss is hardcoded in the implementation.

**Data pre-processing.** In all experiments with images we normalize them to $[-1, 1]$. We rescale MNIST and FashionMNIST images to $32 \times 32$. In all other cases, we rescale images to $64 \times 64$. Note that Fruit360 dataset originally contains $114 \times 114$ images; before rescaling, we add white color padding to make the images have the size $128 \times 128$. Working with Ave, celeba! dataset, we additionally shift each subset $\mathbb{P}_n$ by $(\overline{\mu} - \mu_n)$, i.e., we train the models on the $\lceil \text{CS} \rceil$ baseline. This helps the models to avoid learning the shift.

**Computational complexity**. The most challenging experiments (Ave, celeba! and Handbags, Shoes, Fruit) take about 2-3 days to converge on 4×GPU GTX 1080 ti. Other experiments converge faster.

| Experiment | D | H | N | $G_\xi$ | $v_{\omega_n}$ | $T_{\theta_n}$ | $k_G$ | $k_v$ | $k_T$ | $lr_G$ | $lr_v$ | $lr_T$ | $\ell$ | Total iters | Batch size |
|---|---|---|---|---|---|---|---|---|---|---|---|---|---|---|---|
| Toy 2D | 2 | 2 | 3 | MLP | | MLP | | | 10 | $1 \cdot 10^{-4}$ | $1 \cdot 10^{-3}$ | $1 \cdot 10^{-3}$ | | 12K | 1024 |
| Gaussians | 2-128 | 2-128 | 4 | MLP | | MLP | | | 10 | $1 \cdot 10^{-4}$ | $1 \cdot 10^{-3}$ | $1 \cdot 10^{-3}$ | MSE | 12K | 1024 |
| MNIST 0/1 | 1024 | 16 | 2 | | | | 50 | 50 | 15 | $1 \cdot 10^{-4}$ | $1 \cdot 10^{-4}$ | $1 \cdot 10^{-4}$ | | 60K | |
| FashionMNIST | | | 10 | | | | | | 10 | $1 \cdot 10^{-4}$ | $1 \cdot 10^{-4}$ | $1 \cdot 10^{-4}$ | | 100K | |
| Bags, Shoes, Fruit | 12288 | 128 | 3 | ResNet | | UNet | | | 10 | $3 \cdot 10^{-4}$ | $3 \cdot 10^{-4}$ | $3 \cdot 10^{-4}$ | | 36K | 64 |
| Ave, celeba! | | | 3 | | | | | | 10 | $3 \cdot 10^{-4}$ | $3 \cdot 10^{-4}$ | $3 \cdot 10^{-4}$ | VGG | 60K | |
| Celeba | | | 1 | | | | | | 15 | $1 \cdot 10^{-4}$ | $1 \cdot 10^{-4}$ | $1 \cdot 10^{-4}$ | | 80K | |
| Celeba (fixed $G$) | | | 1 | | | | 0 | | 15 | $1 \cdot 10^{-4}$ | $1 \cdot 10^{-4}$ | $1 \cdot 10^{-4}$ | | 120K | |

Table 3: Hyperparameters that we use in the experiments with our algorithm 1.

## B.3 Hyperparameters (Algorithm 2, Learning Maps to the Barycenter)

After using the main algorithm 1 to train $G_\xi$, we use algorithm 2 to extract the inverse optimal maps $\mathbb{P}_n \to \mathbb{P}_\xi$. We detail the hyperparameters in Table 4 below. In all the cases we use Adam optimizer with the default betas. The column **total iters** show the number of update steps for each $v_{\omega'_n}^{\text{inv}}$.

## B.4 Hyperparameters of competitive [SC$\mathbb{W}_2$B] algorithm

On Ave, celeba! we use [19, Algorithm 1] with $k_3 = 50000$, $k_2 = k_1 = 10$.[5] The optimizer, the learning rates and the generator network are the same as in our algorithm. However, for the potentials

---

[3] `https://github.com/harryliew/WGAN-QC`

[4] `https://github.com/milesial/Pytorch-UNet`

[5] We also tried training their ICNN-based algorithm in our **iterative** manner, i.e., by performing multiple regression updates of the generator instead of the single **variational** update. This provided the same results.

| Experiment | D | N | $v_{\omega_n}$ | $T_{\theta_n}$ | $k_T$ | $lr_v$ | $lr_T$ | Total iters | Batch size |
|---|---|---|---|---|---|---|---|---|---|
| Toy 2D | 2 | 2 | MLP | MLP | 10 | $1 \cdot 10^{-3}$ | $1 \cdot 10^{-3}$ | 10k | 1024 |
| MNIST 0/1 | 1024 | 2 | ResNet | UNet | 10 | $1 \cdot 10^{-4}$ | $1 \cdot 10^{-4}$ | 4k | 64 |
| FashionMNIST | | 10 | | | | | | 4k | |
| Bags, Shoes, Fruit | 12288 | 3 | | | | | | 20K | |
| Ave, celeba! | | 3 | | | | | | 12K | |

Table 4: Hyperparameters that we use in the experiments with algorithm 2

---

**Algorithm 2:** Learning maps from input measures to the learned barycenter $\mathbb{P}_\xi \approx \overline{\mathbb{P}}$ with $\lceil$MM:R$\rceil$ OT solver.

---

**Input** : latent $\mathbb{S}$ and input $\mathbb{P}_1, \dots, \mathbb{P}_N$ measures; pretrained generator $G_\xi : \mathbb{R}^H \to \mathbb{R}^D$ satisfying $G_\xi \sharp \mathbb{S} \approx \overline{\mathbb{P}}$;
 mapping networks $T_{\theta_1'}^{\text{inv}}, \dots, T_{\theta_N'}^{\text{inv}} : \mathbb{R}^D \to \mathbb{R}^D$; potentials $v_{\omega_1'}^{\text{inv}}, \dots, v_{\omega_N'}^{\text{inv}} : \mathbb{R}^D \to \mathbb{R}$;
 number of inner iterations for training transport maps: $K_T$;

**Output** : OT maps satisfying $T_{\theta_n'}^{\text{inv}} \sharp \mathbb{P}_n \approx \mathbb{P}_\xi = (G_\xi \sharp \mathbb{S}) \approx \overline{\mathbb{P}}$;

**repeat**
  **for** $n = 1, 2, \dots, N$ **do**
    Sample batches $Z \sim \mathbb{S}$, $Y \sim \mathbb{P}_n$; $X \leftarrow G_\xi(Z)$;
    $\mathcal{L}_v \leftarrow \frac{1}{|Y|} \sum\limits_{y \in Y} v_{\omega_n'}^{\text{inv}}\big(T_{\theta_n'}^{\text{inv}}(y)\big) - \frac{1}{|X|} \sum\limits_{x \in X} v_{\omega_n'}^{\text{inv}}(x)$;
    Update $\omega_n'$ by using $\frac{\partial \mathcal{L}_v}{\partial \omega_n'}$;
    **for** $k_T = 1, 2, \dots, K_T$ **do**
      Sample batch $Y \sim \mathbb{P}_n$;
      $\mathcal{L}_T = \frac{1}{|Y|} \sum\limits_{y \in Y} \big[\frac{1}{2} \|y - T_{\theta_n'}^{\text{inv}}(y)\|^2 - v_{\omega_n'}^{\text{inv}}\big(T_{\theta_n'}^{\text{inv}}(y)\big)\big]$;
      Update $\theta_n'$ by using $\frac{\partial \mathcal{L}_T}{\partial \theta_n'}$;
**until** *not converged*;

---

(OT solver), we use ICNN architecture as it is required by their method. We use ConvICNN64 [29, Appendix B.1] architecture. For handbags, shoes, fruit (Figure 8), the parameters are the same.

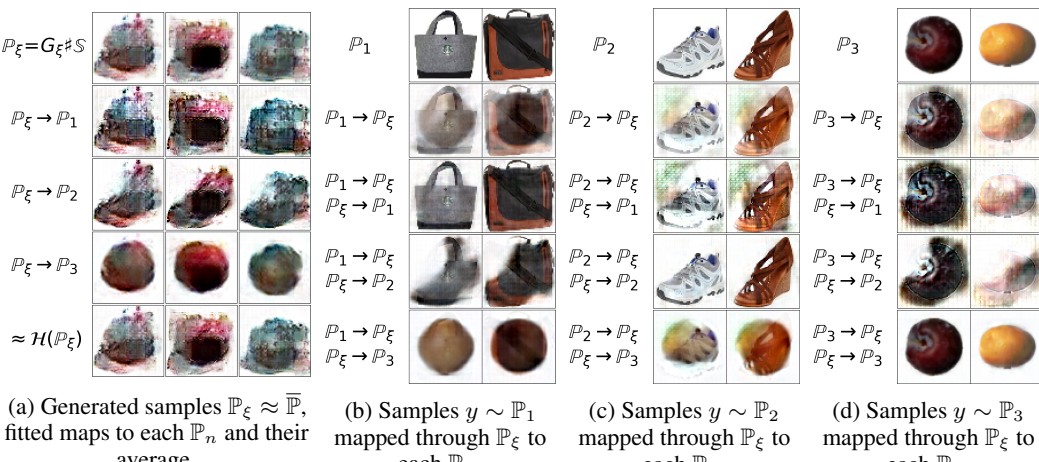

(a) Generated samples $\mathbb{P}_\xi \approx \overline{\mathbb{P}}$, fitted maps to each $\mathbb{P}_n$ and their average.

(b) Samples $y \sim \mathbb{P}_1$ mapped through $\mathbb{P}_\xi$ to each $\mathbb{P}_n$.

(c) Samples $y \sim \mathbb{P}_2$ mapped through $\mathbb{P}_\xi$ to each $\mathbb{P}_n$.

(d) Samples $y \sim \mathbb{P}_3$ mapped through $\mathbb{P}_\xi$ to each $\mathbb{P}_n$.

Figure 8: The barycenter of Handbags, Shoes, Fruit ($64 \times 64$) datasets fitted by **competitive** [SC$\mathbb{W}_2$B].

# C  Additional Experimental Results

## C.1  Toy Experiments

In this section, we provide examples of barycenters computed by our Algorithm for 2D location-scatter cases. To produce the location-scatter population of distributions and compute their ground truth barycenters, we employ the publicly available code[6] of [$\mathbb{CW}_2$B] paper [30]. The hyper-parameters of our Algorithm 1 (learning the barycenter and maps to input measures) and Algorithm 2 ($\lceil$MM:R$\rfloor$ solver, learning the inverse maps) are given in Tables 3 and 4, respectively. For evaluation, we consider two location-scatter populations produced by a rectangle and a swiss-roll respectively [30, §5]. The computed barycenters and maps to/from the input distributions are shown in Figures 9, 10.

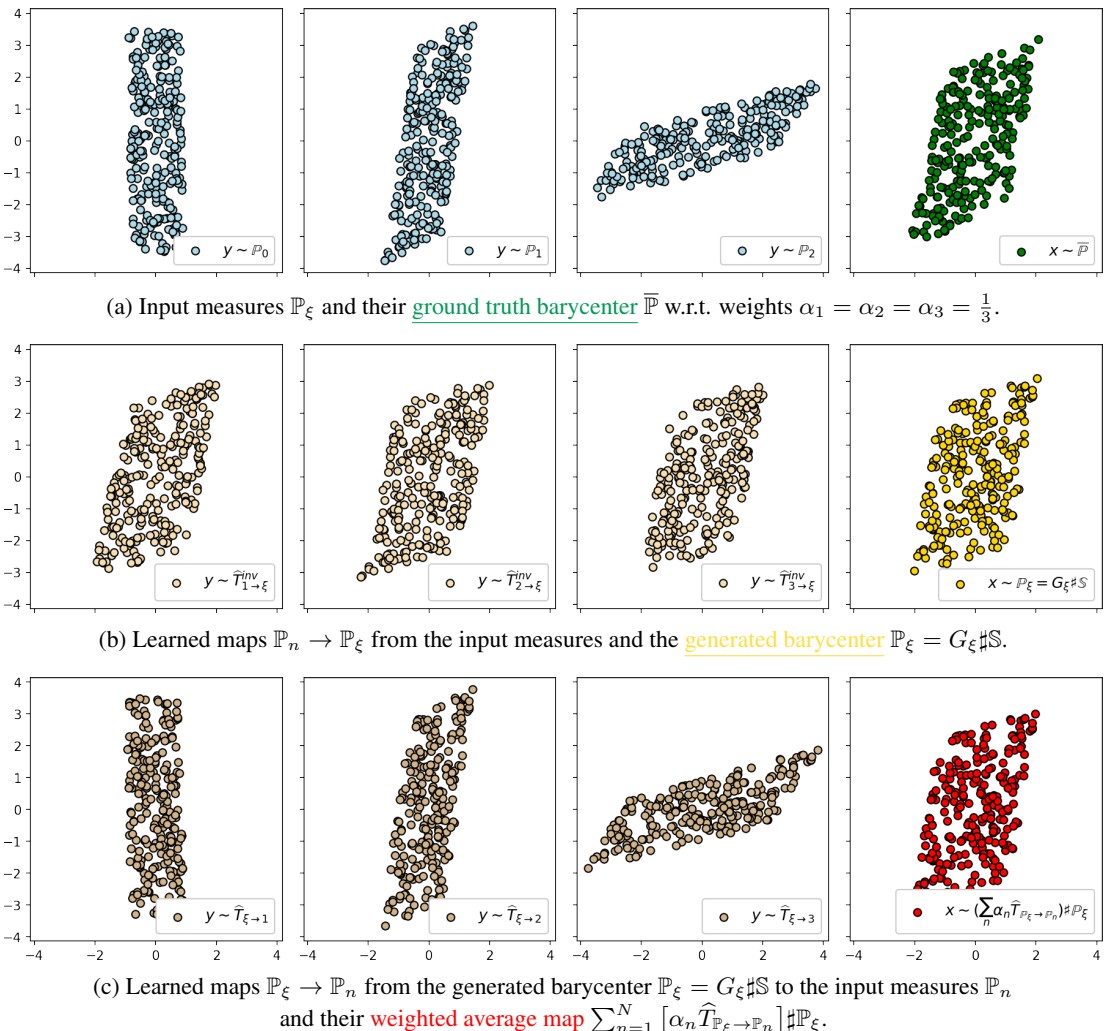

(a) Input measures $\mathbb{P}_\xi$ and their ground truth barycenter $\overline{\mathbb{P}}$ w.r.t. weights $\alpha_1 = \alpha_2 = \alpha_3 = \frac{1}{3}$.

(b) Learned maps $\mathbb{P}_n \to \mathbb{P}_\xi$ from the input measures and the generated barycenter $\mathbb{P}_\xi = G_\xi \sharp \mathbb{S}$.

(c) Learned maps $\mathbb{P}_\xi \to \mathbb{P}_n$ from the generated barycenter $\mathbb{P}_\xi = G_\xi \sharp \mathbb{S}$ to the input measures $\mathbb{P}_n$ and their weighted average map $\sum_{n=1}^{N} \left[ \alpha_n \widehat{T}_{\mathbb{P}_\xi \to \mathbb{P}_n} \right] \sharp \mathbb{P}_\xi$.

Figure 9: The results of applying our algorithm to compute the barycenter of a 2D location-scatter population produced by a rectangle.

## C.2  Location-Scatter Case

Similar to [30, 19], we consider **location-scatter** cases for which the true barycenter can be computed [3, §4]. Let $\mathbb{P}_0 \in \mathcal{P}_{2,\mathrm{ac}}(\mathbb{R}^D)$ and define the following location-scatter family of distributions $\mathcal{F}(\mathbb{P}_0) = \{ f_{S,u} \sharp \mathbb{P}_0 \mid S \in \mathcal{M}_{D \times D}^+, u \in \mathbb{R}^D \}$, where $f_{S,u} : \mathbb{R}^D \to \mathbb{R}^D$ is a linear map $f_{S,u}(x) = Sx + u$ with positive definite matrix $S \in \mathcal{M}_{D \times D}^+$. When $\{\mathbb{P}_n\} \subset \mathcal{F}(\mathbb{P}_0)$, their barycenter $\overline{\mathbb{P}}$ is also an

---

[6]http://github.com/iamalexkorotin/Wasserstein2Barycenters

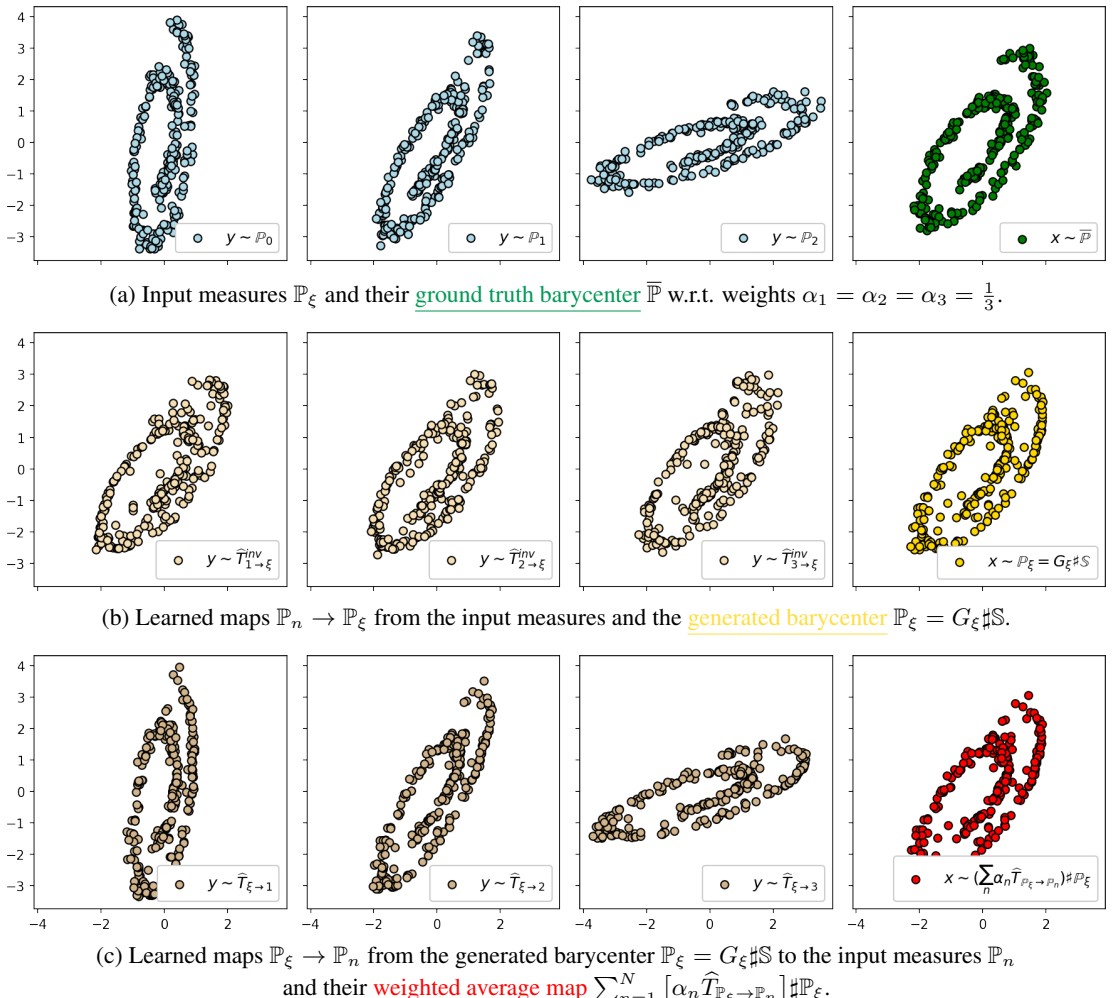

(a) Input measures $\mathbb{P}_\xi$ and their ground truth barycenter $\overline{\mathbb{P}}$ w.r.t. weights $\alpha_1 = \alpha_2 = \alpha_3 = \frac{1}{3}$.

(b) Learned maps $\mathbb{P}_n \to \mathbb{P}_\xi$ from the input measures and the generated barycenter $\mathbb{P}_\xi = G_\xi \sharp \mathbb{S}$.

(c) Learned maps $\mathbb{P}_\xi \to \mathbb{P}_n$ from the generated barycenter $\mathbb{P}_\xi = G_\xi \sharp \mathbb{S}$ to the input measures $\mathbb{P}_n$
and their weighted average map $\sum_{n=1}^N \left[\alpha_n \widehat{T}_{\mathbb{P}_\xi \to \mathbb{P}_n}\right] \sharp \mathbb{P}_\xi$.

Figure 10: The results of applying our algorithm to compute the barycenter of a 2D location-scatter population produced by a Swiss roll.

element of $\mathcal{F}(\mathbb{P}_0)$ and can be computed via fixed-point iterations [3]. We use $N = 4$ measures with weights $(\alpha_1, \ldots, \alpha_4) = (\frac{1}{10}, \frac{2}{10}, \frac{3}{10}, \frac{4}{10})$. We consider two choices for $\mathbb{P}_0$: the $D$-dimensional standard **Gaussian** and the **uniform** distribution on $[-\sqrt{3}, +\sqrt{3}]^D$. By using the publicly available code of [30], we construct $\mathbb{P}_n$ as $f_{S_n^T \Lambda S_n, 0} \sharp \mathbb{P}_0 \in \mathcal{F}(\mathbb{P}_0)$, where $S_n$ is a random rotation matrix and $\Lambda$ is diagonal with entries $[\frac{1}{2}b^0, \frac{1}{2}b^1, \ldots, 2]$ where $b = \sqrt[D-1]{4}$. We quantify the generated barycenter $G_\xi \sharp \mathbb{S}$ with the Bures-Wasserstein Unexplained Variance Percentage [30, §5]:

$$\mathrm{B}\mathbb{W}_2^2\text{-UVP}(G_\xi \sharp \mathbb{S}, \overline{\mathbb{P}}) = 100 \cdot \mathrm{B}\mathbb{W}_2^2(G_\xi \sharp \mathbb{S}, \overline{\mathbb{P}})/\left[\frac{1}{2}\mathrm{Var}(\overline{\mathbb{P}})\right]\%,$$

where $\mathrm{B}\mathbb{W}_2^2(\mathbb{P}, \mathbb{Q}) = \mathbb{W}_2^2\left(\mathcal{N}(\mu_{\mathbb{P}}, \Sigma_{\mathbb{P}}), \mathcal{N}(\mu_{\mathbb{Q}}, \Sigma_{\mathbb{Q}})\right)$ is the Bures-Wasserstein metric and $\mu_{\mathbb{P}}$, $\Sigma_P$ denote mean and covariance of $\mathbb{P}$. The metric admits the closed form [13]. For the trivial baseline prediction $G_{\xi_0}(z) \equiv \mu_{\overline{\mathbb{P}}} \equiv \sum_{n=1}^N \alpha_n \mu_{\mathbb{P}_n}$ the metric value is 100%. We denote this baseline as $\lfloor C \rfloor$.

| Method | D=2 | 4 | 8 | 16 | 32 | 64 | 128 | Method | D=2 | 4 | 8 | 16 | 32 | 64 | 128 |
|---|---|---|---|---|---|---|---|---|---|---|---|---|---|---|---|
| $\lfloor C \rfloor$ | 100 | 100 | 100 | 100 | 100 | 100 | 100 | $\lfloor C \rfloor$ | 100 | 100 | 100 | 100 | 100 | 100 | 100 |
| [SC$\mathbb{W}_2$B] | 0.07 | 0.09 | 0.16 | 0.28 | 0.43 | 0.59 | 1.28 | [SC$\mathbb{W}_2$B] | 0.12 | 0.10 | 0.19 | 0.29 | 0.46 | 0.6 | 1.38 |
| **Ours** | 0.01 | 0.02 | 0.01 | 0.08 | 0.11 | 0.23 | 0.38 | **Ours** | 0.04 | 0.06 | 0.06 | 0.08 | 0.11 | 0.27 | 0.46 |

Table 5: Comparison of $\mathrm{B}\mathbb{W}_2^2$-UVP$\downarrow$ (%) in the location-scatter cases:
$\mathbb{P}_0 = \mathcal{N}(0, I_D)$ on the **left** and $\mathbb{P}_0 = \mathrm{Uniform}\left([-\sqrt{3}, +\sqrt{3}]^D\right)$ on the **right**.

The results of our algorithm 1 and $[\text{SCW}_2\text{B}]$ adapted from [30, Table 1] are given in Table 5. Both algorithms work well in the location-scatter cases and provide $\text{B}\mathbb{W}_2$-UVP $< 2\%$ in dimension 128.

## C.3 Generative Modeling

Analogously to [19], we evaluate our algorithm when $N = 1$. In this case, the minimizer of (5) is the measure $\mathbb{P}_1$ itself, i.e., $\overline{\mathbb{P}} = \mathbb{P}_1$. As the result, our algorithm 1 works as a usual generative model, i.e., it fits data $\mathbb{P}_1$ by a generator $G_\xi$. For experiments, we use CelebA $64 \times 64$ dataset. Generated images $G_\xi(z)$ and $\widehat{T}_{\mathbb{P}_\xi \to \mathbb{P}_1}\big(G_\xi(z)\big)$ are shown in Figure 11a.

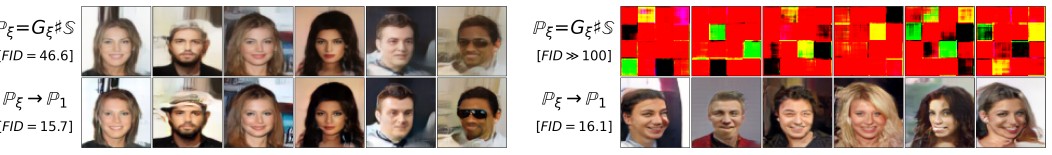

(a) Generator $G_\xi$ training enabled ($K_G > 0$).   (b) Generator $G_\xi$ training disabled ($K_G = 0$).

Figure 11: Images generated by our algorithm 1 serving as a generative model. The 1st line shows samples from $G_\xi \sharp \mathbb{S} \approx \mathbb{P}_1$, the 2nd line shows estimated OT map $\widehat{T}_{\mathbb{P}_\xi \to \mathbb{P}_1}$ from $G_\xi \sharp \mathbb{S}$ to $\mathbb{P}_1$ which further improves generated images.

In Table 6, we provide FID for generated images. For comparison, we include FID for ICNN-based $[\text{SCW}_2\text{B}]$, and WGAN-QC [35]. FID scores are adapted from [29, §4.5]. Note that for $N = 1$, $[\text{SCW}_2\text{B}]$ is reduced to the OT solver by [40] used as the loss for generative models, a setup tested in [29, Figure 3a]. Serving as a generative model when $N = 1$, our algorithm 1 performs comparably to WGAN-QC and *drastically* outperforms ICNN-based $[\text{SCW}_2\text{B}]$.

| *Method* | | *FID↓* |
|---|---|---|
| $[\text{SCW}_2B]$ | $G_\xi(z)$ | 90.2 |
| | $\widehat{T}_{\mathbb{P}_\xi \to \mathbb{P}_1}\big(G_\xi(z)\big)$ | 89.8 |
| WGAN-QC | $G_\xi(z)$ | 14.4 |
| **Ours** | $G_\xi(z)$ | 46.6 |
| | $\widehat{T}_{\mathbb{P}_\xi \to \mathbb{P}_1}\big(G_\xi(z)\big)$ | 15.7 |
| **Ours** (fixed $G_\xi$) | $G_\xi(z)$ | N/A |
| | $\widehat{T}_{\mathbb{P}_\xi \to \mathbb{P}_1}\big(G_\xi(z)\big)$ | 16.1 |

Table 6: FID scores of generated faces.

**Fixed generator.** For $N = 1$, the fixed point approach §4.1 converges in only one step since operator $\mathcal{H}$ immediately maps $G_\xi \sharp \mathbb{S}$ to $\mathbb{P}_1$. As a result, in our algorithm 1, *exclusively* when $N = 1$, we can fix generator $G_\xi$ and train only OT map $T_{\theta_1}$ from $G_\xi \sharp \mathbb{S}$ to data measure $\mathbb{P}_1$ and related potential $v_{\omega_1}$. As a sanity check, we conduct such an experiment with randomly initialized generator network $G_\xi$. The results are given in Figure 11b, the FID is included in Table 6. Our algorithm performs well even *without generator training* at all.

## C.4 Barycenters of MNIST Digits and FashionMNIST Classes

Similar to [19, Figure 6], we provide qualitative results of our algorithm applied to computing the barycenter of two MNIST classes of digits $0, 1$. The barycenter w.r.t. weights $(\frac{1}{2}, \frac{1}{2})$ computed by our algorithm is shown in Figure 12. We also consider a more complex FashionMNIST [66] dataset. Here we compute the barycenter of 10 classes w.r.t. weights $(\frac{1}{10}, \dots, \frac{1}{10})$. The results are given in Figures 13 and Figure 14.

Due to (6), each barycenter images are an average (in pixel space) of certain images from the input measure. In all the Figures, the produced barycenter images satisfy this property. The maps to input measures are visually good. The approximate fixed point operator $\mathcal{H}(\mathbb{P}_\xi)$ is almost the identity as expected (the method converged).

## C.5 Additional Results

In Figure 16, we visualize maps between Ave, Celeba! subsets through the learned barycenter. In Figure 15, we provide additional qualitative results for computing barycenters of Handbags, Shoes, Fruit360 datasets.

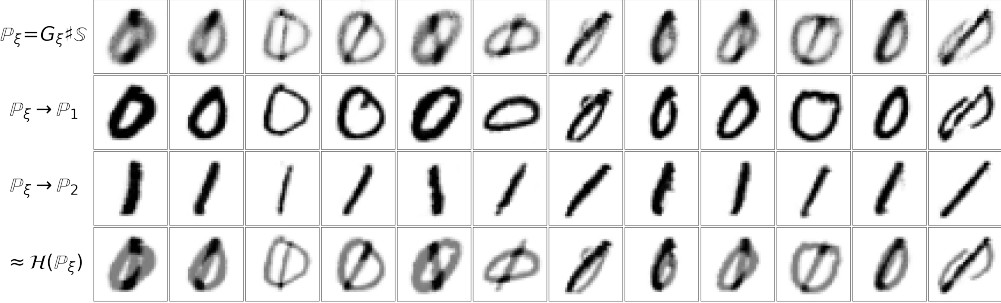

(a) The barycenter $\mathbb{P}_\xi$ and maps to input measures $\mathbb{P}_n$.

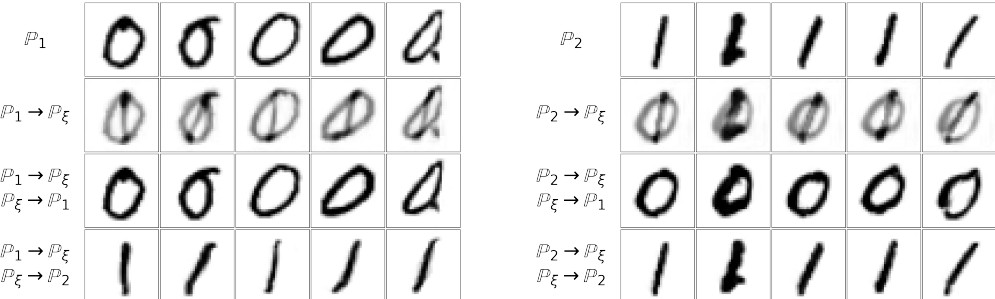

(b) Samples from $\mathbb{P}_1$ mapped through $\mathbb{P}_\xi$ to each $\mathbb{P}_n$.    (c) Samples from $\mathbb{P}_2$ mapped through $\mathbb{P}_\xi$ to each $\mathbb{P}_n$.

Figure 12: The barycenter of MNIST digit classes 0/1 learned by Algorithm 1.

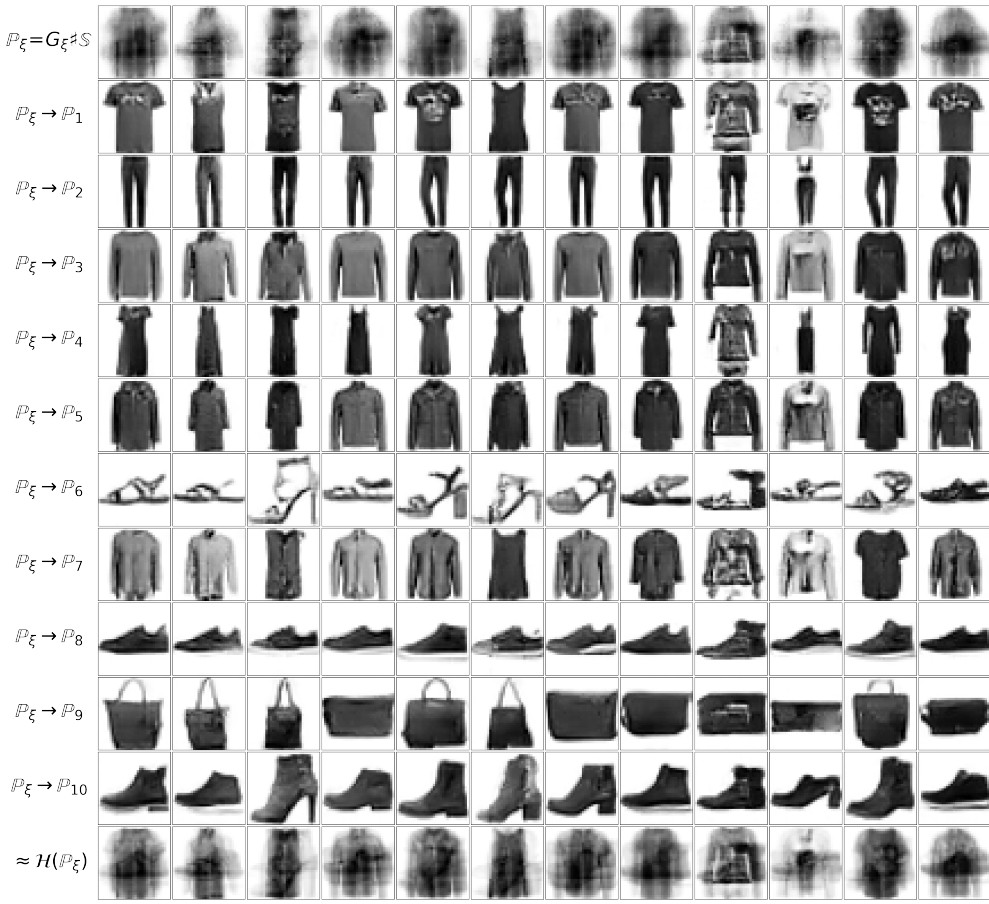

Figure 13: The barycenter and maps to input measures estimated by our method on 10 FashionMNIST classes (32×32). The 1st line shows generated samples from $\mathbb{P}_\xi = G_\xi \sharp \mathbb{S} \approx \overline{\mathbb{P}}$. Each of 10 next lines shows estimated optimal maps $\widehat{T}_{\mathbb{P}_\xi \to \mathbb{P}_n}$ to measures $\mathbb{P}_n$. The last line shows average $\left[ \sum_{n=1}^N \alpha_n \widehat{T}_{\mathbb{P}_\xi \to \mathbb{P}_n} \right] \sharp \mathbb{P}_\xi$.

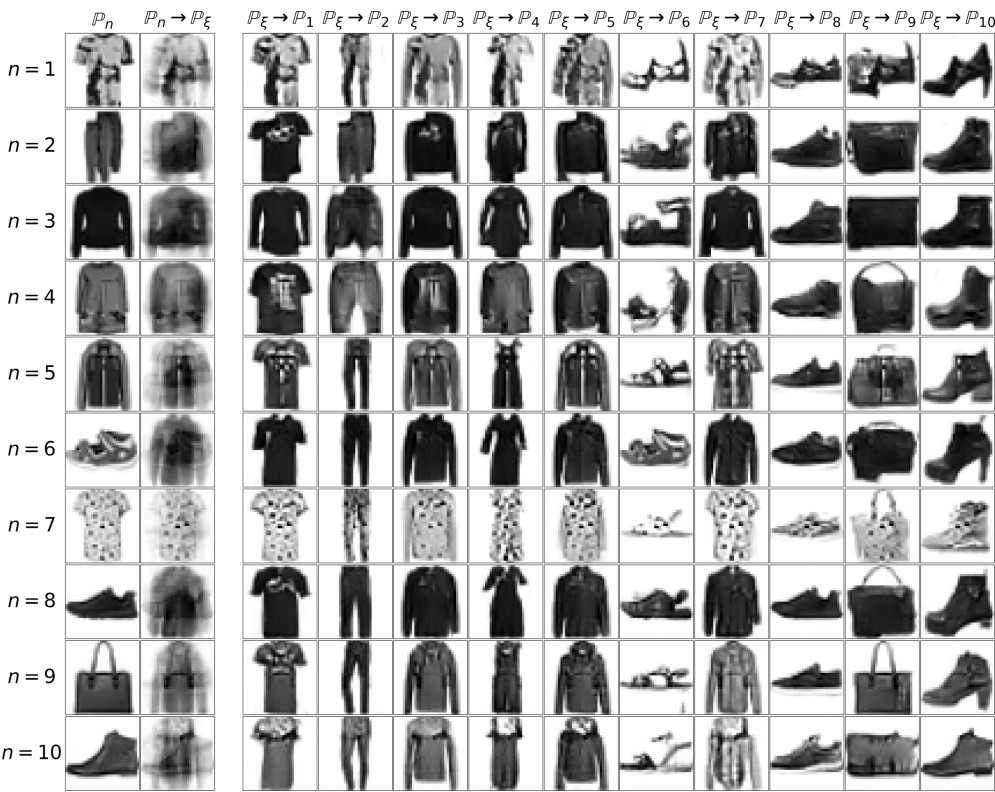

Figure 14: Maps between FashionMNIST classes through the learned barycenter. The 1st images in each $n$-th column shows a sample from $\mathbb{P}_n$. The 2nd columns maps these samples to the barycenter. Each next column shows how the maps from the barycenter to the input classes $\mathbb{P}_n$.

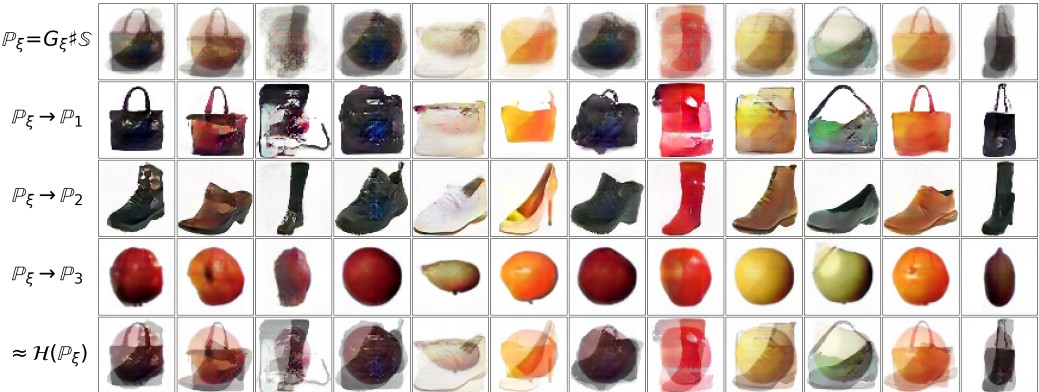

(a) Generated samples $\mathbb{P}_\xi \approx \overline{\mathbb{P}}$, fitted maps to each $\mathbb{P}_n$ and their average.

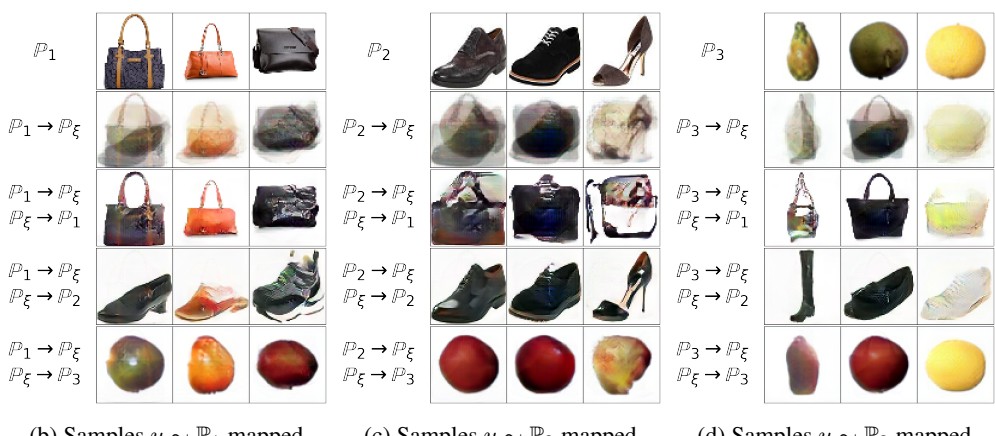

(b) Samples $y \sim \mathbb{P}_1$ mapped through $\mathbb{P}_\xi$ to each $\mathbb{P}_n$.

(c) Samples $y \sim \mathbb{P}_2$ mapped through $\mathbb{P}_\xi$ to each $\mathbb{P}_n$.

(d) Samples $y \sim \mathbb{P}_3$ mapped through $\mathbb{P}_\xi$ to each $\mathbb{P}_n$.

Figure 15: The barycenter of Handbags, Shoes, Fruit ($64 \times 64$) datasets fitted by our algorithm 1.

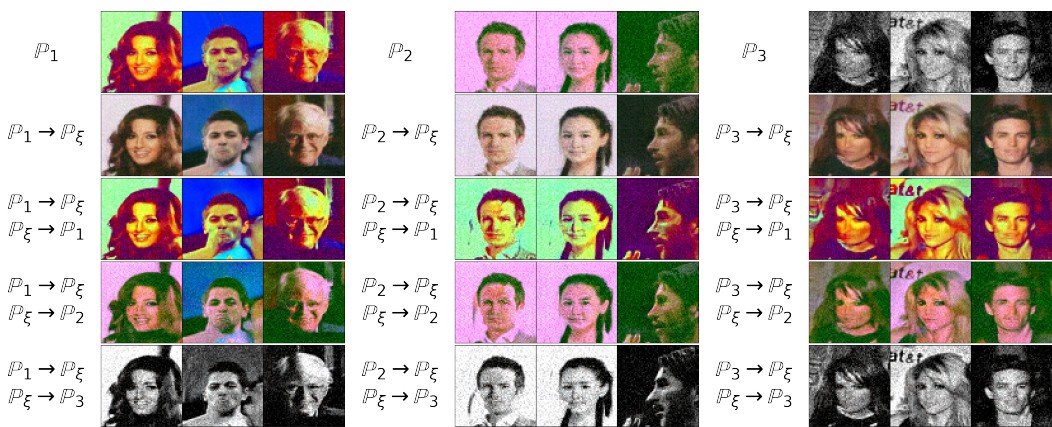

(a) Samples $y \sim \mathbb{P}_1$ mapped through $\mathbb{P}_\xi$ to each $\mathbb{P}_n$.

(b) Samples $y \sim \mathbb{P}_2$ mapped through $\mathbb{P}_\xi$ to each $\mathbb{P}_n$.

(c) Samples $y \sim \mathbb{P}_3$ mapped through $\mathbb{P}_\xi$ to each $\mathbb{P}_n$.

Figure 16: Maps between subsets of Ave, celeba! dataset through the barycenter learned by our algorithm 1.