# OpenReview forum: "Wasserstein Iterative Networks for Barycenter Estimation"
_NeurIPS.cc/2022/Conference — NeurIPS 2022 Accept_

### Official Review · Reviewer_3V2X · 2022-06-30

**Rating:** 6
**Confidence:** 4
**Soundness:** 2 fair
**Presentation:** 3 good
**Contribution:** 3 good

**Summary:**

This paper proposes to learn the Wasserstein barycenter through the fixed point iteration method. They interchangeably 1) solve the OT map between the barycenter candidate and each marginal distribution, 2) update the barycenter candidate by fixed-point iteration, then regress a generator to fit the barycenter candidate. It also develops a CelebA barycenter dataset that has a known barycenter solution.

**Questions:**

The paper claims they cannot obtain the inverse map: marginal distribution -> barycenter during training, but the inverse map can be recovered by $x- \nabla v(x)$ if using eq 7 to solve the optimal map. This is discussed at the end of section 2 in [W2 solver benchmark paper](https://openreview.net/pdf?id=CI0T_3l-n1). Why not use this inverse map as a baseline?

In Figure 5, SCWB is particularly bad for the map corresponding to $P_3$. This is somehow weird. Do authors have clue why this happens?

For the ablation study, I think it would be helpful if the authors can add some experiments with $k_G=1$. Thus we can be more clear about whether SCWB has worse performance because ICNN is not expressive enough.

**Limitations:**

see weakness

**Strengths And Weaknesses:**

Strengths:

It provides a nice interpolation between the gradient updates of the least square regression to fixed point barycenter update and the variational barycenter algorithms.

It gives theoretical guidance to generating probability measures with known Wasserstein barycenter.

It obtains advantageous performance compared to the competitive SCWB algorithm.

Weaknesses:

As at the beginning of section 4.1, the fixed point doesn't promise to be the barycenter, and the entire sequence may not converge, which may cause numerical issues. These two challenges would question the foundation of the proposed fixed-point iteration solver.

The paper claims to tackle challenge c, but the algorithm still needs to compute N maps. So it doesn't decrease the calculation burden, at least when compared with SCWB.

It seems to me that computational time is rarely mentioned in the paper. For example, the paper uses $k_G=50$, and also a large number of iterations $k_v=50$, while the corresponding parameter is only 1 and 10 for SCWB. I conjecture the computational time of this method is several times larger than SCWB.

---

> ### Author Response · Authors · 2022-08-02
> **Response to Reviewer 3V2X**
>
> Thank you for your valuable feedback. Please find above (in our reply to all the Reviewers) the answers to your comments common with other reviews. Please below find below our answers to your questions that do not overlap with those of other Reviewers.
>
> **(1) Inverse map.**
>
> As the reviewer suggests, the inverse map $P_{n}\rightarrow P_{\xi}$ indeed may be obtained from the gradient $\nabla v_{n}(x)$ of the trained potential $v_{n}$. However, such a strategy suffers from the **gradient deviation** issue which is revealed in the related Wasserstein-2 benchmark paper [24] --- see the end of their Section 2. Namely, the recovered potential $\widehat{v}$ may be close to the true potential $v^{\star}$, i.e., $\hat{v}\approx v^{\star}$, but this does not imply that $\nabla\hat{v}\approx \nabla v^{\star}$. This is a serious practical issue --- see Figure 5c and nearby discussion in [24]. Thus, we learn the inverse OT maps separately (directly as a neural network, not as a gradient of the potential) after training the main algorithm.
>
> **(2) The map corresponding to $P_{3}\rightarrow P_{\xi}$ in $\lceil\mbox{SCWB}\rfloor$ algorithm is bad (Figure 3c).**
>
> The ground truth map from the barycenter $\overline{P}$ to the marginal measure $P_{3}$ is \textit{nearly} a decolorization, i.e., it has a very small Lipschitz constant. Thus, the inverse ground truth map $P_{3}\rightarrow P_{\xi}$ has a very high Lipschitz constant and is hard to learn. Due to being very restrictive, it seems like ICNNs simply struggle to learn this map whereas our method does not have such an issue.
>
> **Concluding remarks**. Please respond to our post to let us know if the clarifications above suitably address your concerns about our work. We are happy to address any remaining points during the discussion phase; if the responses above are sufficient, we kindly ask that you consider raising your score.

---

> > ### Comment · Reviewer_3V2X · 2022-08-06
> > **Reply after rebuttal**
> >
> > Hi! Authors,
> >
> > Thank you for your reply! There is only one thing left that I don't feel comfortable with. That is: this paper uses $k_G=50, k_v=50$, while the corresponding parameter is only 1 and 10 for SCWB. I think this is somehow unfair because
> >
> > 1) at least $k_v, k_T$ in this paper vs $k_1,k_2$ in SCWB are both for the OT solver, you should keep two pairs of parameters the same;
> >
> > 2) even though SCWB uses the iteration for the generator as $k_G=1$, to solve (8) with stochastic gradient descent, you can always increase $k_G$ to a larger number for SCWB.
> >
> > I think the experiment should be based on a fair comparison with comparable iteration parameters, i.e. 1) making $k_1 = k_T , k_2 = k_v,$ 2) increasing the iteration parameter of the generator of SCWB from 1 to $k_G$ in this paper.
> >
> > Considering the limited time, I encourage the authors to add one experiment meeting these two requirements and have FID comparison. I would like to change my score if this concern is solved.

---

> > > ### Author Response · Authors · 2022-08-09
> > > **Response to Reviewer 3V2X - further evaluation of ICNN-based method**
> > >
> > > Dear reviewer, we have conducted the experiment which you have suggested. Namely, we trained ICNN-based solver in an *iterative* manner, $k_{G}=50$ generator updates, $k_{1}=k_{T}=10$ transport map updates, $k_{2}=k_{v}=50$ updates of the potential. The qualitative results of this experiment on two datasets (Handbag-shoes-fruit and our constructed Ave, Celeba!) are in the **Experiments** folder of the revised submission's supplementary material.
> > >
> > > **Results on Handbag, shoes, fruit.** (*experiments/handbag-shoes-fruit* folder). Here we perform only the qualitative analysis as the ground truth barycenter is not known. The obtained results look comparable to the ones we reported in the paper for the ICNN solver (Figure 8 in Appendix, $k_2 = k_1 = 10$).
> > >
> > > **Results on Ave, celeba!** (*experiments/ave-celeba* folder). The generated faces are blurry and look comparable to Figure 4b for ICNN method with $k_2 = k_1 = 10$. Note that the map corresponding to $P_{3}\rightarrow P_\xi$ is particularly bad. This matches the initial experiment with SCWB about which you asked a question in your initial review. The FID scores of generated images $G(z)$ and mapped images $\sum\alpha_{n}T_{n}(G(z))$ are **161.7** and **160.4** respectively, which is even higher than those we originally reported for $k_2 = k_1 = 10$ (**156.3**/**152.1**). We think that this change in FID just due to the variance of attempts. FID $\gg$ 100 is already a large value and $\pm 10$ FID at this level is not important.
> > >
> > > To conclude, for ICNNs method there is not notable difference between iterative/variational training. In our view, the main cornerstone is the ICNN parameterization which is too restrictive.
> > >
> > > We will add the conducted experiment to the final version of the paper. Please respond to our post to let us know if the clarifications above suitably address your concerns about our work.

---

> > > > ### Comment · Reviewer_3V2X · 2022-08-09
> > > > **reply**
> > > >
> > > > Hi! Thank you for your efforts. I'd like to increase my score. Please distribute your rebuttal discussions into the revision. Especially, please add a discussion of the inverse map. Now I still see in line 251, you have "Our algorithm does not compute them (inverse map)", this can be misleading to readers. You actually already calculate/approximate them with your method, it's just practically you cannot recover the inverse map well.

---

### Official Review · Reviewer_NEXn · 2022-07-02

**Rating:** 7
**Confidence:** 3
**Soundness:** 4 excellent
**Presentation:** 4 excellent
**Contribution:** 4 excellent

**Summary:**

Computing the Wasserstein barycenter between continuous measures is an important problem. In this paper, authors propose a method based on a fixed-point approach to compute the Wasserstein barycenter with arbitrary neural networks. Moreover, they introduce a way to create datasets to compare to quality of the obtained barycenter and compare their method to a ICNN based method on the "ave, celeba!" dataset, which they generated using their method.

**Questions:**

Some related works are not cited. In [1], a method is proposed to compute continuous Wasserstein barycenter for abitrary costs. I believe it would be worth also comparing the method against their method.

[2,3] are also a missing reference on the methods used to approximate Wasserstein barycenters.

The method seems pretty heavily computationally (since it requires to compute N OT maps at each iteration and solve a regression problem), but I guess it is also the case of the other method which is compared with. I would be curious to see some comparison of the training time between the methods.


[1] Chi, Jinjin, et al. "Variational Wasserstein Barycenters with c-Cyclical Monotonicity." arXiv preprint arXiv:2110.11707 (2021).

[2] Backhoff-Veraguas, Julio, et al. "Stochastic gradient descent in Wasserstein space." arXiv preprint arXiv:2201.04232 (2022).

[3] Daaloul, Chiheb, et al. "Sampling from the wasserstein barycenter." arXiv preprint arXiv:2105.01706 (2021).

**Limitations:**

Yes

**Strengths And Weaknesses:**


This paper propose to use generative model into the fixed-point approach previously introduced. This allows to use NNs to parametrize the barycenter and to obtain better results than other papers. Moreover, they introduce a way to construct benchmark datasets to compare the efficiency of the method to compute Wasserstein barycenters. This is a very nice contribution which can be very helpful to compare methods.

Strengths:
- Fixed-Point approach with neural networks and the regression approach.
- A new way to construct a benchmark dataset to compare methods on Wasserstein barycenters


Weaknesses:
- Comparison with only one method. A more complete comparison with the different methods on the newly introduced dataset could be nice.
- Still some open questions on whether the fixed-point approach approximates the barycenter.

---

> ### Author Response · Authors · 2022-08-02
> **Response to Reviewer NEXn**
>
> Thank you for your valuable feedback. Please find above (in our reply to all the Reviewers) the answers to your comments common with other reviews. Please below find below our answers to your questions that do not overlap with those of other Reviewers.
>
> **(1) Comparisons to additional barycenter algorithms.**
>
> We thank you for pointing to relevant works "*Variational Wasserstein Barycenter*" $\lfloor \mbox{VWB}\rceil$, "*Sampling from the Wasserstein Barycenter*" $\lfloor \mbox{SWB}\rceil$, and "*Stochastic gradient descent in Wasserstein space*" $\lfloor \mbox{SGDW}\rceil$ paper. We will include the citations to the final version of the paper. $\lfloor \mbox{SGDW}\rceil$ is an interesting theoretical paper, while $\lfloor \mbox{VWB}\rceil$, $\lfloor \mbox{SWB}\rceil$ provide computational procedures for the barycenters. However, we think that performing comparisons with the latter two is also not needed, as we explain below.
>
> In the Gaussian (or location-scatter) case, most existing methods score very small errors ($<2\%$ in the BW-UVP metric which is standard for this task). More precisely, see Table 2 in $\lfloor \mbox{VWB}\rceil$ paper, Figure 5 in $\lfloor \mbox{SCWB}\rceil$, or Tables 1-3 in $\lfloor \mbox{CWB}\rceil$ paper [25]. As we show in Appendix C.1, our method also easily achieves such low errors.
>
> Neither $\lfloor \mbox{VWB}\rceil$ nor $\lfloor \mbox{SWB}\rceil$ provides available source code. Moreover, $\lfloor \mbox{VWB}\rceil$ and $\lfloor \mbox{SWB}\rceil$ consider only small-dimensional experiments, not on image spaces (the focus of our paper). The method $\lfloor \mbox{VWB}\rceil$ has a large amount of tunable hyperparameters and technical details (see their Section 3.2 and Algorithm 1), which are not trivial to setup  in a different setting that is not considered in their paper, i.e., image spaces. The paper $\lfloor \mbox{SWB}\rceil$ differs from the topic of our paper because it is a particle-based, not  neural method: methods involving space discretization are known to be notably outperformed by neural methods in high-dimensions in OT-related tasks --- see, e.g., Tables 1-3 in [25] or Figures 1, 3 in [37].
>
> *Due to the reasons above, we think that comparisons with these barycenter methods are not needed since they are not suitable to be tested on high-dimensional domains such as the images space.*
>
> **Technical remark**. The revised paper yet does not contain the citations for the above mentioned papers. This is needed to avoid the shift in the numbering of references during the discussion period. We will add the citations to the final version of the paper.
>
> Please respond to our post to let us know if the clarifications above suitably address your concerns about our work. We are happy to address any remaining points during the discussion phase; if the responses above are sufficient, we kindly ask that you consider raising your score.

---

> > ### Comment · Reviewer_NEXn · 2022-08-05
> > **Response to authors**
> >
> > Thank you for your answers.
> >
> > The authors addressed my concerns. Therefore, I will improve my rating to accept.

---

### Official Review · Reviewer_xsZZ · 2022-07-11

**Rating:** 6
**Confidence:** 3
**Soundness:** 3 good
**Presentation:** 2 fair
**Contribution:** 3 good

**Summary:**

Authors propose a methodology for estimation of wasserstein barycenter in continuous setup. Authors argue that it beats the state of the art. In the passing authors conceive a new way of transforming/splitting datasets leading to new 'ave, celeba' dataset which might be useful in the future for benchmarking in OT tasks.

**Questions:**

Why do you only compare against SCW2B? I understand that this is the state of the art but by only comparing against it, and given that there are multiple components to your methodology, it is hard to identify where the gains come from. I suggest to whenever further experiment that allow us to debug where the gains come from. What are the advantages to each of the components in section 4?

I strongly suggests to add a toy example. Is there any reason for why this should be not necessary? For example, one could try to find the barycenter of some simple continuous gaussian data. Adding such toy example (which suitable baselines, even if they don't correspond to the SOTA SCW2B) would help 'debug' and interpret the computational gains.

Methods related: on many parts it is mentioned that ICNN don't work so well but at the same time the congruent functions are computed with ICNN. Could you make this distinction more explicit in the text (why for dataset creation it makes sense to use ICNN while they don't work so well for barycenter estimation?)

**Limitations:**

To me understanding authors correctly discuss limitations. But lack of sufficient interpretability is a substantial limitation of this work, as it is submitted at this point.

**Strengths And Weaknesses:**

The main strengths are that it is a thoughtful paper on a relevant topic that keeps demanding new methods, and where the proposed methodology is valuable because the 'tricks' are novel, and because they lead to better results. It is clear to me that there is a large amount of work involved on this project.

The new dataset is also a major strength of this paper, and I hope authors will work in order to make it easily publicly available for future work. Although the math is rather simple, it is a strength of the paper that the use of math leads to quite valuable methodological tricks.

The main weakness is that although there are certainly quite a few clever methodological nuggets, altogether it is very hard to identify how everything is put together to lead to a better algorithm (see for example questions below). If that is not clear enough then the community has reasons to be skeptical about the evaluation. This is mostly an interpretability issue and I hope it can be addressed.

This requires some work in two related aspects: 1) although writing is clear, what is less clear to me are the different components of the proposed algorithm.  Figure 2 is beautiful but it is not clear enough. Here I missed the usual Algorithm box. I believe authors should try to polish the methods section (4) so it is crystal clear what the contributions are. I would have love to have a more thorough discussion than 'it works better empirically'. Hopefully, we would see a clear justification to every design choice made in section 4.

---

> ### Author Response · Authors · 2022-08-02
> **Response to Reviewer xsZZ**
>
> Thank you for your valuable feedback. Please find above (in our reply to all the Reviewers) the answers to your comments common with other reviews. Please find below our answers to your questions that do not overlap with those of other Reviewers.
>
> **(1) Future public availability of the dataset and code.**
>
> We will make all our code and the constructed Ave, celeba! dataset publicly available.
>
> **(2) Toy examples.**
>
> Following your suggestion, we added toy 2D examples to Appendix D of the revised version of the paper.
>
> **(3) Algorithm box.**
>
> The algorithm box with the detailed training procedure is given in Appendix B.2. In the final version, we will move it to the main text, as per reviewer's request.
>
> **(4) Why ICNNs are not good for computing the barycenter and but suitable to create the Ave, celeba! dataset?**
>
> As we explained in the general answer to all the reviewers, the ICNNs do not work well in large-scale real-world tasks presumably due to their low expressiveness (compared to common convolutional architectures), i.e., they are too restrictive. However, to create a dataset with the known ground truth barycenter, it is necessary to use a **convex parameterization** for the potentials $\psi_{n}$. Unfortunately, to our knowledge, ICNNs are **the only way** to achieve this, so we have to cope with their suboptimal expressiveness when creating the dataset.
>
> **Concluding remarks**. Please respond to our post to let us know if the clarifications above suitably address your concerns about our work. We are happy to address any remaining points during the discussion phase; if the responses above are sufficient, we kindly ask that you consider raising your score.

---

### Official Review · Reviewer_idDE · 2022-07-14

**Rating:** 5
**Confidence:** 4
**Soundness:** 2 fair
**Presentation:** 3 good
**Contribution:** 2 fair

**Summary:**

The article proposes an iterative procedure to compute Wasserstein barycenters, the parametrised barycenter is modelled as a latent distribution pushed forward by a neural network.

The authors study the relationship with existing methods and use their proposal to build a benchmark dataset for barycentric computation

**Questions:**

How can the author justify the general applicability of their method given that the experiments only consider purposed-built datasets?

**Ethics Review Area:**

["I don’t know"]

**Limitations:**

the limitations of this work are identified in the manuscript

**Strengths And Weaknesses:**

The paper does a great job in the literature review, where it puts its method in context and compares it with other techniques in terms of their advantages and disadvantages.

The theoretical treatment of the contribution is well explained, the algorithm consists in iteratively applying a neural operator to a (latent) density according the known fixed point barycentric method by [3]. The proposal is validated in the celebrity dataset.

In my opinion, the weakest point of the article is the experimental validation. Though the proposal is sound, only two sets of experiments are shown: i) evaluation of a synthetic dataset built upon the proposed method, and ii) other synthetically-generated dataset. These experiments hardly validate the idea of a general method for barycenter computation, since there is no evidence that the proposed method operates beyond datasets built in the specific manner mentioned by the authors. This issue is identified in the manuscript, in Sec 7 (lines >295) where the authors acknowledge that their dataset only cater for specific (dispersion) properties - given this observation, it was clear that the experimental validation should've included richer datasets.

This weakness in the contribution is further clarified when the authors identify a possible application in MRI (Sec 7), where datasets have limited overlap. Perhaps a stronger experimental validation could include real world datasets of that kind.

Though well written in general, the article could benefit from a proofreader

---

> ### Author Response · Authors · 2022-08-02
> **Response to Reviewer idDE**
>
> Thank you for your valuable feedback! Please find above (in our reply to all the Reviewers) the answers to your comments common with other reviews. Please find below our answers to your questions that do not overlap with those of other Reviewers.
>
> **(1) There is no evidence that the proposed method operates beyond datasets built in the specific manner.**
>
> In the paper, to validate our proposed algorithm, we conduct two main types of experiments.
>
> **(a) Quantitative experiments.** Here we consider only datasets with the known ground truth; otherwise, quantitative analysis of the recovered barycenter is impossible. We consider the constructed Ave, celeba! dataset (one contribution of this paper, Section 6.1) and the location-scatter case (Appendix C.1). As we point in lines 46-51 and 56-59, to our knowledge, these cases are essentially **the only** ones with the known ground truth barycenter. It is possible but unlikely that we miss some other datasets with known ground truth barycenters; if this is the case please kindly let us know which dataset we have overlooked.
>
> **(b) Qualitative experiments.** Here we consider various image datasets (Section 6.2 and Appendices C.3 and C.4, MNIST, FashionMNIST, Handbags/Shoes/Fruit) without the ground truth barycenters. Thus, we can only qualitatively analyze the results. In this experiment, the datasets are **not synthetic** and *they are not built in the specific manner* for the barycenter task. Our results on Figures 6, 10-13 are the **evidence** that the proposed method remains effective beyond synthetic datasets.
>
> *Additionally, we would like to emphasize that our experimental evaluation is already more comprehensive than those of related papers. We consider more challenging setups (both synthetic Ave, celeba! and non-synthetic Handbag/Shoes/Fruit). For example, the images that we work with are RGB and their resolution is $2\times 2$ times bigger than those in the experiments of $\lfloor\mbox{SCWB}\rceil$. This yields $2\times 2\times 3=12$ times bigger dimension of the ambient space.*
>
> **(2) General applicability.**
>
>  As we write in lines 273-278, our algorithm could help to aggregate data from multiple sites and overcome the distributional shift issue across sites. We agree with the reviewer position that it would be nice to test the algorithm, e.g., using MRI data (line 274). Studying particular application such as MRI is a promising future research avenue, which would require overcoming considerable technical issues (proper data collection for MRI, handling data privacy issues, etc.) which are too domain-specific and are beyond the scope of the current paper, which focuses on methodology of a large-scale barycenter algorithm and a generic way to generate barycenter datasets, plus the new barycenter Ave, celeba! dataset.
>
> **Concluding remarks**. Please respond to our post to let us know if the clarifications above suitably address your concerns about our work. We are happy to address any remaining points during the discussion phase; if the responses above are sufficient, we kindly ask that you consider raising your score.

---

> > ### Comment · Reviewer_xsZZ · 2022-08-07
> > **Thank you**
> >
> > Thanks for my response. I'll keep my score

---

> ### Author Response · Authors · 2022-08-08
> **Looking forward to your final feedback**
>
> Dear Reviewer idDE ,
>
> We thank you for your review and appreciate your time reviewing our paper.
>
> The end of the rebuttal phase is approaching. We would be grateful if we could hear your feedback regarding our answers to the reviews. We are happy to address any remaining points during the remaining period.
>
> Thanks in advance,
>
> Paper5431 authors

---

> > ### Comment · Reviewer_idDE · 2022-08-09
> > **thanks**
> >
> > Dear Authors,
> >
> > many thanks for the detailed general and particular responses. I appreciate the explanation of the authors and thus I raise my score.

---

### Author Response · Authors · 2022-08-02
**General Response (part 1)**

Dear reviewers, we thank you for your insightful comments and interesting questions! We are glad that you positively highlight our theoretical insights (Reviewer 3V2X), clear writing (Reviewer xsZZ) and literature review (Reviewer idDE), consider our methodology valuable (Reviewer xsZZ), and agree that the constructed benchmark is helpful for comparing future barycenter methods (Reviewer NeXn). Please find the answers to your shared questions below.

**(1) Which element provides the improved performance compared to the SCWB?** (Reviewers xsZZ, 3V2X)

Recall that compared to the $\lceil \text{SCWB}\rfloor$, our algorithm has two major changes:

**(a)** The $\lceil \text{MM:R}\rfloor$ solver with more expressive neural networks replaces the ICNN-based solver to compute OT maps;

**(b)** Fixed points updates (Figure 2b, regression) replace variational updates (8).

The **major gain** comes from **(a)**. The fact that ICNNs-based methods are not suitable for large-scale problems has already been demonstrated in the benchmark paper [24]. Specifically, we refer to their Figure 5a (ICNN-based solver) and Figure 5d ($\lceil \text{MM:R}\rfloor$ solver). There the authors use $\lceil \text{MM:R}\rfloor$ and ICNN-based solvers to estimate the loss for GANs. This is the same as computing the barycenter of a single input measure $P_{1}$ with the variational approach (see additionally Section 3.4 of $\lceil \text{SCWB}\rfloor$ [15]), i.e., $k_{G}=1$. Their experiment demonstrates that $\lceil \text{MM:R}\rfloor$ outperforms ICNN-based solvers by a **large** margin in the image generation task (FID **18.8** of $\lceil \text{MM:R}\rfloor$, FID **90** of an ICNN-based solver).

More importantly, we refer the reviewers to the recent "*Variational Wasserstein gradient flow*" paper (ICML 2022), co-authored by the author (Amirhossein Taghvaei) of $\lceil \text{SCWB}\rfloor$ barycenter method and ICNN-based OT solvers [35,57]. There it is directly shown that regular (non-ICNN) parameterization leads to notably improved performance in an OT-related task --- see their Figure 3a and 4a.

Next we show **(b)** also leads to considerable improvement. In Appendix C.2, we conduct an experiment testing the **iterative** method ($k_{G}\gg 1$) in the case with one marginal measure $P_1$, i.e., simply training the generative model for data. The FID scores of the generated barycenter $P_{\xi}$ and the transport map $P_{\xi}\rightarrow P_{1}$ (Figure 9a of our paper) are **46.6/15.7** respectively. On the other hand, we refer the reviewers to Figure 5d of [24]. There the authors test $\lceil \text{MM:R}\rfloor$ but with **variational** updates (this is exactly equivalent to $k_{G}=1$) for the generator on the same CelebA faces dataset. Their reported FID scores are  **58.2/18.8** which is bigger. Thus, **(b)** also provides noticeable gain, though smaller than **(a)**.

**(2) Comparisons to other alternative algorithms beyond $\lceil \text{SCWB}\rfloor$.** (Reviewers xsZZ, NEXn)

We compare only to $\lceil \text{SCWB}\rfloor$ because it is the **state-of-the-art**, and, to our knowledge, it is the **only** algorithm which has been shown to work at the large-scale such as in the high-dimensional space of images --- see their Figures 6, 7. Besides, $\lceil \text{SCWB}\rfloor$ uses a generative model (as our algorithm does) which makes the comparison more fair.

**(3) Fixed points are not necessarily the barycenters/convergence to the barycenter** (Reviewers 3V2X, NEXn)

This is not a problem which is specific to our iterative algorithm -- the existing **variational approaches**, e.g., $\lfloor \text{SCWB}\rceil$, also **suffer from** exactly **the same issue**. Namely, the fixed point for their iterative algorithms can also be a **local minima** for the variational algorithm. Indeed, in every **fixed** point $P_{\xi}$ we have $\sum_{n}\alpha_{n} T_{n}(x)= x$. Thus, according to our Lemma 1, it follows that $\frac{\partial}{\partial \xi}=0$ *in the variational algorithm*, i.e., *the generator gets stuck in the fixed point as well.*

---

> ### Author Response · Authors · 2022-08-02
> **General Response (Part 2)**
>
> **(4) Computational time.** (Reviewers 3V2X, NEXn)
>
> We report the computational time of our method in lines 568-569 of Appendix B.2. Training our method on the most challenging **Handbags, Shoes, Fruit!** dataset takes 3 days on 4 GPUs GTX 1080ti. $\lfloor \text{SCWB}\rceil$  takes roughly one day to converge to the reported FID value **150+** and then FID simply stops improving. Thus, despite the fact it converges faster, it converges to a significantly worse solution (our FID is $\approx$ **50**, see Table 1). The inference time for generating the barycenter images is the same for both methods since we use the same ResNet generator network.
>
> In the Gaussian case, the comparisons are not representative as all the methods converge in the matter of minutes.
>
> *Thus, despite the fact that the training time of our method is higher than $\lceil\text{SCWB}\rfloor$, it achieves much better results. We demonstrate this quantitatively and qualitatively in our experiments.*
>
> **Revision**. We have uploaded an updated version of the paper. The new version contains new *Appendix D with toy 2D experiments* (pages 25-26, lines 616+). The newly added content is highlighted with the **blue** color.
>
> **Concluding remarks.** Please respond to this post to let us know if the clarifications above suitably address your concerns about our work. We are happy to address any remaining points during the discussion phase; if the responses above are sufficient, we kindly ask that you consider raising your score.

---

### Meta-Review · Area_Chair_QiSK · 2022-08-25

**Recommendation:** Accept
**Confidence:** Certain

**Metareview:**

This paper proposes a new iterative method for Wasserstein barycenter based on a generator parametrization of the barycenter and fixed point method that alternates learning the generator and learning  the OT maps from barycenter to measures. The work is empirical and lacks theory but proposes a new image benchmark on celeba for barycenter evaluation. Reviewers were positive about the paper. Accept.

**Award:**

No

---

### Decision · Program_Chairs · 2022-09-14

Accept